# Multi-kingdom gut microbiota analyses define COVID-19 severity and post-acute COVID-19 syndrome

Qin Liu [1,2,3,4,8], Qi Su [1,2,3,4,8], Fen Zhang [1,2,3,4], Hein M. Tun[1,3,5], Joyce Wing Yan Mak[1,2,3,4], Grace Chung-Yan Lui [2,6], Susanna So Shan Ng[2], Jessica Y. L. Ching[1,2,3,4], Amy Li[2,3,4], Wenqi Lu[1,2,3,4], Chenyu Liu[1,2,3,4], Chun Pan Cheung[1,2,3,4], David S. C. Hui [2,6], Paul K. S. Chan [4,7], Francis Ka Leung Chan[1,2,3,4] & Siew C. Ng [1,2,3,4] ✉

Our knowledge of the role of the gut microbiome in acute coronavirus disease 2019 (COVID-19) and post-acute COVID-19 is rapidly increasing, whereas little is known regarding the contribution of multi-kingdom microbiota and host-microbial interactions to COVID-19 severity and consequences. Herein, we perform an integrated analysis using 296 fecal metagenomes, 79 fecal metabolomics, viral load in 1378 respiratory tract samples, and clinical features of 133 COVID-19 patients prospectively followed for up to 6 months. Metagenomic-based clustering identifies two robust ecological clusters (hereafter referred to as Clusters 1 and 2), of which Cluster 1 is significantly associated with severe COVID-19 and the development of post-acute COVID-19 syndrome. Significant differences between clusters could be explained by both multi-kingdom ecological drivers (bacteria, fungi, and viruses) and host factors with a good predictive value and an area under the curve (AUC) of 0.98. A model combining host and microbial factors could predict the duration of respiratory viral shedding with 82.1% accuracy (error ± 3 days). These results highlight the potential utility of host phenotype and multi-kingdom microbiota profiling as a prognostic tool for patients with COVID-19.

The coronavirus disease-2019 (COVID-19) pandemic has affected over 500 million people and killed 6 million people worldwide. Identifying predictors of disease severity and deterioration is a priority to guide clinicians and policymakers for better clinical management, resource allocation, and long-term management of COVID-19 patients. Several lines of evidence, such as replication of severe acute respiratory syndrome coronavirus 2 (SARS-CoV-2) in human enterocytes[1–3], detection of viruses in fecal samples[4,5], and altered gut microbiota composition, including the increased abundance of opportunistic pathogens and reduced abundance of beneficial symbionts in the gut of patients with COVID-19 suggest involvements of the gastrointestinal (GI) tract[6–9].

[1]Microbiota I-Center (MagIC), Hong Kong SAR, China. [2]Department of Medicine and Therapeutics, Faculty of Medicine, The Chinese University of Hong Kong, Hong Kong, Hong Kong SAR, China. [3]Li Ka Shing Institute of Health Sciences, State Key Laboratory of Digestive Disease, Institute of Digestive Disease, Faculty of Medicine, The Chinese University of Hong Kong, Hong Kong SAR, China. [4]Center for Gut Microbiota Research, Faculty of Medicine, The Chinese University of Hong Kong, Hong Kong SAR, China. [5]The Jockey Club School of Public Health and Primary Care, Faculty of Medicine, The Chinese University of Hong Kong, Hong Kong SAR, China. [6]Stanley Ho Centre for Emerging Infectious Diseases, Faculty of Medicine, The Chinese University of Hong Kong, Shatin, Hong Kong. [7]Department of Microbiology, Faculty of Medicine, The Chinese University of Hong Kong, Hong Kong SAR, China. [8]These authors contributed equally: Qin Liu, Qi Su. ✉e-mail: siewchienng@cuhk.edu.hk

Recent studies have shown that gut dysbiosis is linked to the severity of COVID-19 and persistent complications months after disease resolution[7,8,10]. Patients with severe disease exhibit elevated plasma concentrations of inflammatory cytokines and markers, including interleukin-6 (IL-6), IL-8, and IL-10, lactate dehydrogenase (LDH), and C-reactive protein (CRP), reflecting immune responses and tissue damages after SARS-CoV-2 infection[11,12]. Among hospitalized COVID-19 patients, gut microbiota composition is also associated with blood inflammatory markers[7], and the lack of short-chain fatty acids and L-isoleucine biosynthesis in the gut microbiome are correlated with disease severity[13].

In addition to bacteria, the human gut is home to a vast number of viruses and fungi that regulate host homeostasis, physiological processes, and the assembly of co-residing gut bacteria, which could potentially play an important role in the pathophysiological mechanisms that determine COVID-19 outcomes. Since the therapeutic potential for COVID-19 patients includes approaches to inhibit, activate, or modulate immune function, it is essential to define these characteristics related to clinical features in a well-defined patient cohort. We hypothesized that microbial interaction networks may improve our understanding of the pathophysiology and long-term consequences of COVID-19. Here, using an unsupervised classification approach based on fecal metagenomic profiling and blood inflammatory markers, we demonstrated that integrative microbiomes from a multi-kingdom network provide a novel framework for understanding disease complications and have potential applications in risk stratification and prognostication of COVID-19 cases.

## Results

### Multi-omics analysis reflects disease severity and clinical symptoms in COVID-19 patients

We included 133 hospitalized patients with COVID-19 in three hospitals in Hong Kong between 13 March 2020 and 27 January 2021. We assessed viral RNA levels in nasopharyngeal swabs and fecal samples using reverse transcription quantitative real-time PCR (RT-qPCR). We also assessed plasma cytokine and chemokine levels and leukocyte profiles in freshly isolated peripheral blood mononuclear cells (PBMCs). We also analyzed the gut microbiome composition (bacteria, viruses, and fungi) in 296 serial fecal samples collected at up to three longitudinal time-points from admission to six months after virus clearance using shotgun metagenomic sequencing and assessed the metabolomics of 79 fecal samples at admission (Figs. 1, 2A). In total, 296 stool samples were sequenced, generating an average of 6.9 Gbp per sample.

The gut multi-biome (bacteria, fungi, and viruses) profile at admission was integrated using an unsupervised weighted similarity network fusion (WSNF) approach[14]. Weighting was assigned according to the total number of observed taxa present in a particular biome, with filtering based on a prevalence of at least 5% across the patient cohort[14]; virome (732 species) > bacteriome (242 species) > mycobiome (12 species) observed across 133 patients. By subjecting multi-biome data to this non-supervised similarity network fusion approach, fecal samples were divided into two distinct patient clusters based on the microbiota matrix: 47.4% of patients in WSNF-Cluster 1 (n = 63), and 52.6% (n = 70) in WSNF-Cluster 2 (Fig. 2B).

We next compared microbial profiles between clusters (adjusted for age, gender and comorbidity). The multi-biome composition of patients in Cluster 1 was characterized by a predominance of bacteria (*Ruminococcus gnavus, Klebsiella quasipneumoniae*), fungi (*Aspergillus flavus, Candida glabrata, Candida albicans*), and viruses (*Mycobacterium phage MyraDee, Pseudomonas virus Pf1*) (Fig. 2C, MaAsLin2, q < 0.1, Supplementary Data 1). They also exhibited significantly lower multi-biome diversity (Wilcoxon test, p = 0.029, Supplementary Fig. 1A) than those in Cluster 2. Principal Coordinates Analysis (PCoA) of multi-biome composition revealed a significant difference between the two clusters using permutational multivariate analysis of variance (PERMANOVA) (p < 0.001, Supplementary Fig. 1B and Supplementary Data 2).

We found that patients belonging to Cluster 1 exhibited more symptoms such as diarrhea and chills (twofold increased risk), fever, and cough (1.3-fold increased risk; Chi-square, p value < 0.001, q < 0.1) than those in Cluster 2 at admission (Fig. 2D). They were also characterized by a higher viral load (Fig. 2E), greater disease severity (Fig. 2F), increased CRP levels (Fig. 2G), elevated C–X–C motif chemokine 10 (CXCL10) (Fig. 2H), longer duration of viral positivity in upper respiratory tract samples (Supplementary Fig. 2A) and a higher rate of viral positivity in fecal samples (Supplementary Fig. 2B) than those in Cluster 2. We also tested the viral load in fecal samples and found no significant differences between the two clusters (Supplementary Fig. 2C). Demographics and comorbidities were comparable between Cluster 1 and Cluster 2, except that patients within Cluster 1 were 9.2 years older than those in Cluster 2 (Table 1). Patients in Cluster 1 primarily comprised subjects with severe COVID-19 who exhibited more clinical signs (Fig. 2F, D) and these subjects presented with higher plasma CRP and chemokine levels, including CXCL10, which is known to be involved in leukocyte trafficking[15,16]. These observations indicate that gut multi-biome profiles of COVID-19 patients at admission are associated with disease severity, and Cluster 1 was defined as representing patients with more severe disease.

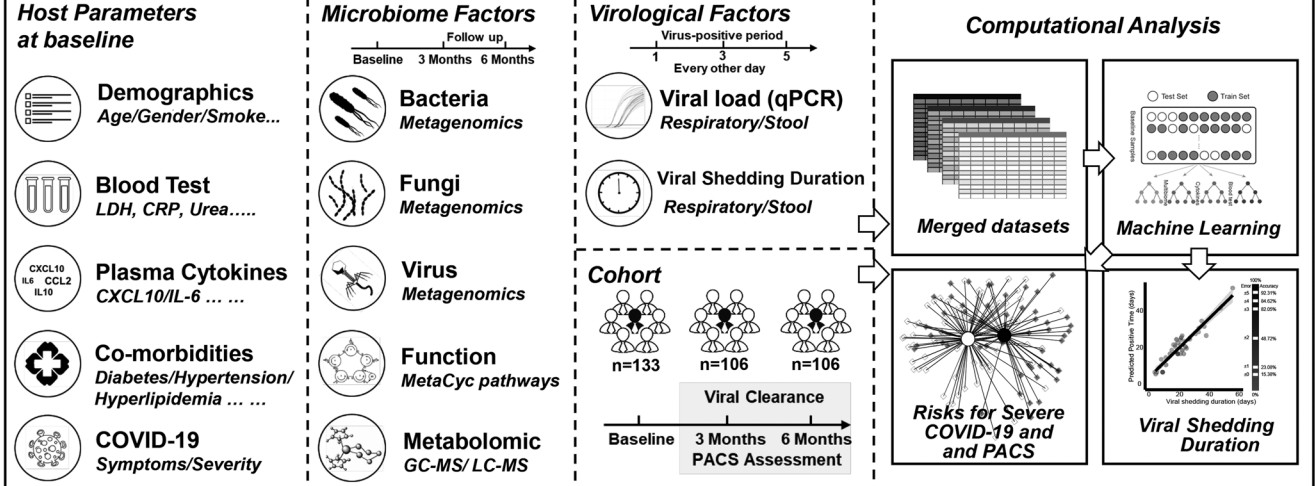

**Fig. 1 | Schematic diagram of study design.** An integrated approach to investigate the prognostic roles of multi-kingdom microbiome, host parameters, and virological factors in COVID-19 outcomes and consequences.

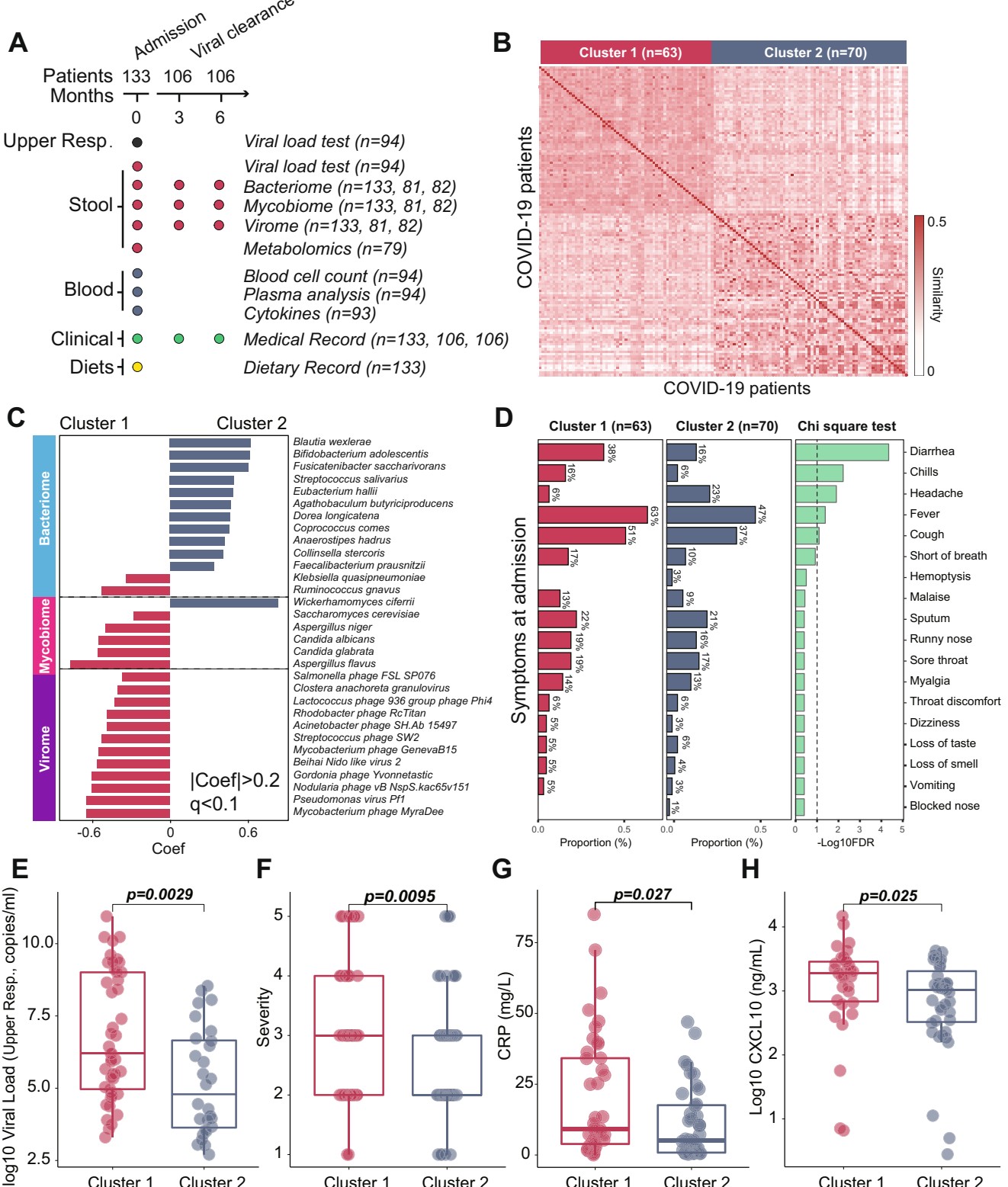

We explored the functional profiling of microbiome signatures in the two clusters and identified cluster-specific functional signatures (Supplementary Fig. 1C and Supplementary Data 3). For functional annotation, we used the Human Microbiome Project Unified Metabolic Analysis Network 3 (HUMAnN3) pipeline, which maps reads to functionally annotated organism genomes and uses a translated search to align unmapped reads to UniRef90 protein clusters[17]. Amongst all microbiome functionalities, urea cycle, L-isoleucine degradation I, and

L-arginine degradation II were enriched in Cluster 1 (Supplementary Fig. 1C, $q < 0.1$, fold change >2). Elevated blood urea nitrogen (BUN) levels have been reported to be associated with critical illness and mortality in COVID-19 patients and are predictive of poor clinical outcomes[15,18]. We found that blood urea levels were strongly associated with the microbiome urea cycle pathway and were higher in COVID-19 patients with severe disease (Supplementary Figs. 1D, E, 3A). Next, we investigated how specific microbiome species were

**Fig. 2 | Integration of gut multi-biome data through weighted similarity network fusion (WSNF) approach. A** Schematic overview of the study design, depicting the total number of samples and participants from whom data were available. **B** Heatmap illustrating pairwise patient WSNF similarity scores stratified by spectral clustering (Cluster 1, n = 63; Cluster 2, n = 70) according to integrated multi-biome profiles, derived from n = 133 biologically independent samples. **C** MaAslin2 analysis of observed clusters illustrating discriminant taxa at baseline (FDR-adjusted q < 0.1). **D** Symptoms of COVID-19 patients between two identified patient clusters. The proportion of diarrhea, chills, headache, fever, and cough in Cluster 1 were significantly higher than in Cluster 2 (Cluster 1, n = 63; Cluster 2, n = 70) (Chi-square test with one degree of freedom, Benjamini–Hochberg correction, p < 0.05; q < 0.1). **E** Comparison of viral load (copies/mL). **F** Severity of disease. **G** C-reactive protein (CRP) concentration. **H** CXCL levels in two identified patient clusters. In **E**–**H**, the two-sided Wilcoxon rank-sum test was used to check the differences between the two clusters. Boxplot lower and upper hinges correspond to the first and third quartiles, upper and lower whiskers represent the highest and lowest values within 1.5 times the interquartile range, and the horizontal line represents the median.

associated with elevated BUN levels in patients with severe COVID-19. The relative abundances of the urea cycle pathway and K01940 in the urea cycle were significantly higher in Cluster 1. Furthermore, we found a marked increase in K01940 (argininosuccinate synthase, the key enzyme in the urea cycle pathway, Supplementary Fig. 3B) in the severe cluster (Supplementary Fig. 3C), which was predominantly driven by *Klebsiella* species such as *Klebsiella quasipneumonia*, *Klebsiella pneumoniae*, and *Klebsiella variicola* (Supplementary Fig. 3D), by comparing subclass pathways and microbial contributors (quantifying gene presence and abundance in a species-stratified manner). High urea level is commonly an indication of kidney dysfunction. However, in our cohort, there was no significant difference in other blood markers of liver and kidney functions (total protein, alkaline phosphatase (ALP), alanine transaminase (ALT), creatinine, Supplementary Data 4, 5), except blood urea. Given the signatures that correlate with disease deterioration, gut-derived uremic toxins in the systemic circulation might be one of the explanations for the marked increase in urea in severe COVID-19 patients. Enriched L-isoleucine degradation I and L-arginine degradation II, and decreased L-isoleucine biosynthesis IV, as well as pyruvate fermentation to acetate and lactate II, were further verified by metabolomics sequencing and correlation analysis (Supplementary Fig. 4).

## Integrative microbiome signatures and post-acute COVID-19 syndrome (PACS)

An exaggerated immune system response, cell damage, or physiological consequences of COVID-19 may contribute to the persistent and prolonged effects after acute COVID-19, known as post-acute COVID-19 syndrome (PACS). The exact pathophysiological mechanisms underlying PACS remain unclear[10,19,20]. By following the gut microbiome dynamics of patients with COVID-19 from admission until six months after viral clearance, we explored microbiome composition (bacteria, viruses, and fungi) at admission and the association with the development of PACS. Although older age was recognized in Cluster 1, there were no significant differences in the age of patients with PACS after six months between the two clusters. For α-diversity based on the Shannon index, we found higher values at 3 months than in baseline samples, but there was no significant increase in the diversity of the microbiota at 6 months (Supplementary Fig. 5A, B). Within Cluster 1 and Cluster 2, there was no significant difference in the gut microbiome composition at admission and follow-up samples at 3 and 6 months (Supplementary Fig. 5C, D, p > 0.05) within each cluster, suggesting that the gut microbiome profile was stable over time. We further assessed whether there were temporal changes in patients without PACS in Cluster 2. The multi-microbiome exhibited stable microbiome profiles from baseline to as long as 6 months of follow-up (Supplementary Fig. 5E, 5F), indicating the persistent impact of SARS-COV-2 infection on the gut microbiome. After 6 months, patients in Cluster 1 exhibited significantly different gut microbiota compositions than those in Cluster 2 (Fig. 3A). The bacteria diversity in Cluster 1 was significantly lower than that in Cluster 2 (Fig. 3A, p = 0.0061). Cluster 1 was characterized by an increase in opportunistic pathogenic bacterial species, including *Erysipelatoclostridium ramosum*[21,22], *Clostridium bolteae*[23], and *Clostridium innocuum*[24] at 6 months (adjusted for age, gender, and comorbidities, Fig. 3B). Significantly more patients within Cluster 1 (84 vs. 44%; FDR <0.1, Chi-square test) developed symptoms of PACS, including insomnia (23 vs. 2%; FDR <0.1), anxiety (28 vs. 7%; FDR <0.1) and poor memory (37% vs. 5%; FDR <0.1), compared with those in Cluster 2 (Fig. 3C).

## Host-microbial factors predict the duration of respiratory viral shedding in COVID-19

We next incorporated host parameters (patient demographics, blood parameters, and cytokine levels) with the microbiome analysis of baseline samples. Using random forest modeling of both host factors and microbiome signatures and a stratified ten-fold cross-validation (Fig. 4A), this model could differentiate Cluster 1 and Cluster 2 with an area-under receiver operator curve (AUROC) of 0.94 (Fig. 4B and Supplementary Data 6). In contrast, a model that incorporated patient demographics (i.e., age, gender, and comorbidities), blood parameters (CRP and LDH), cytokines (i.e., CXCL10, IL-1b, and IL-10), and microbiome analysis alone achieved an AUC of 0.53, 0.60, 0.61, and 0.84, respectively, in differentiating the two clusters (Supplementary Data 6). Patients in Cluster 1 were characterized by more advanced age, higher LDH levels, a greater relative abundance of *Candida albicans* and *Pseudomonas virus Pf1*, and lower relative abundance of *Bifidobacterium adolescentis* and *Faecalibacterium prausnitzii* (Fig. 4C–G). We next evaluated the sub-model performance from the top five to the top 20 and found that using the top 11 achieved the best performance based on this model. With further limitation to the top 11 factors on the random forest, our model achieved an AUC of 0.98, differentiating between the two clusters. These 11 factors included host factors (age, viral load, blood LDH, CRP, and CXCL10 levels), bacteria

**Table 1 | Comparison of clinical characteristics in COVID-19 patients stratified by the integrative multi-kingdom microbiome**

|  | Overall | Cluster 1 | Cluster 2 | *p* |
|---|---|---|---|---|
| Patients, n | 133 | 63 | 70 |  |
| Female, n (%) | 59 (44.4%) | 30 (47.6%) | 29 (41.4%) | 0.207 |
| Age, years (IQR) | 42.2 (26–59) | 47.1 (28.5–63) | 37.9 (20.5–55) | 0.005 |
| Non-smokers, n (%) | 72 (54.1%) | 34 (53.9%) | 38 (54.3%) | 0.402 |
| Presence of any comorbidities, n (%) | 52 (39.1%) | 25 (39.7%) | 27 (38.6%) | 0.393 |
| Hypertension | 28 (21.1%) | 11 (17.5%) | 18 (25.7%) | 0.099 |
| Hyperlipidaemia | 25 (18.8%) | 11 (17.5%) | 14 (20.0%) | 0.335 |
| Diabetes mellitus | 10 (7.5%) | 4 (6.3%) | 6 (8.6%) | 0.287 |
| Length of stay in the hospital, days (IQR) | 21.4 (13–28) | 23.8 (15.5–29) | 19.22 (10–23.75) | 0.025 |
| Severity of COVID-19, n (%) |  |  |  | 0.010 |
| Asymptomatic | 7 (5.3%) | 2 (3.2%) | 5 (7.1%) | 0.153 |
| Mild | 52 (39.1%) | 19 (30.2%) | 33 (47.1%) | 0.007 |
| Moderate | 47 (35.3%) | 25 (39.7%) | 22 (31.4%) | 0.114 |
| Severe | 15 (11.3%) | 9 (14.3%) | 6 (8.6%) | 0.080 |
| Critical | 12 (9.0%) | 8 (12.7%) | 4 (5.7%) | 0.016 |

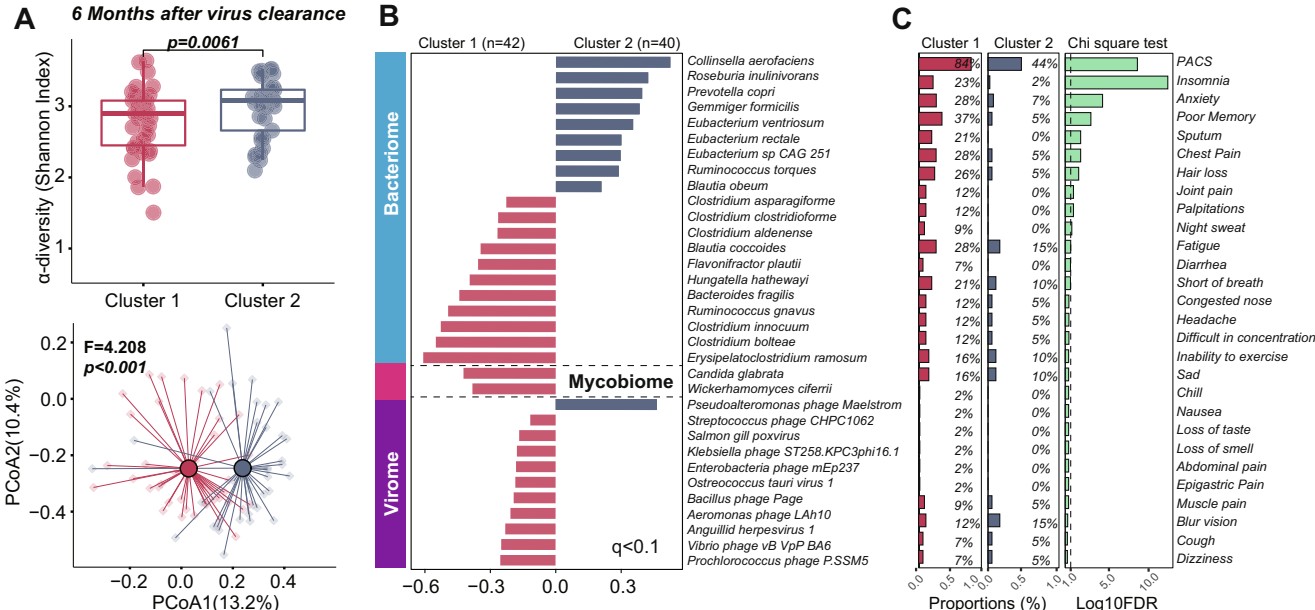

**Fig. 3 | Prognostic roles of gut integrative microbiomes for post-acute COVID-19 syndrome. A** Comparison of α-diversity (Shannon diversity index, two-sided Wilcoxon rank-sum test, $p = 0.029$) of patients at 6 months after viral clearance between two identified patient clusters (Cluster 1 at 6 months, $n = 42$; Cluster 2 at 6 months, $n = 36$) and principal coordinate analysis (PCoA) of gut multi-biome of patients at 6 months after virus clearance based on Bray–Curtis dissimilarity illustrates two patient clusters (PERMANOVA: Adonis test). The line in the boxplot indicates the median value. Box plots lower and upper hinges correspond to the first and third quartiles and upper and lower whiskers represent the highest and lowest values within 1.5 times the interquartile range. **B** MaAslin analysis of observed clusters illustrating discriminant taxa at 6 months after virus clearance. **C** Comparison of post-acute symptoms of COVID-19 patients in two clusters (Chi-square test with one degree of freedom, Benjamini–Hochberg correction, $p < 0.05$; $q < 0.1$).

(*Bifidobacterium adolescentis*, *Faecalibacterium prausnitzii*, and *Blautia wexlerae*), fungi (*Candida albicans* and *Aspergillus niger*) and virus (*Pseudomonas virus Pf1*) composition (Supplementary Data 7). These data suggest that a combination of host and microbial factors provides the most accurate discrimination ability for defining subjects with severe COVID-19.

To explore whether the integration of clinical data with deep microbiome profiling could predict the duration of viral shedding in COVID-19 patients, we tested 1,378 samples from the upper respiratory tract (sputum and nasopharyngeal samples) for the presence of SARS-CoV-2 virus using RT-qPCR every two days for each patient. The median duration of viral shedding (based on positive RT-qPCR) was 21.1 days (IQR 14.5–24.5, range 4-56) after the onset of initial symptoms. We used a random forest analysis of ensembled datasets (demographics, blood tests, cytokines, and multi-biome) to predict the duration of viral shedding in an individual patient. Using a discovery cohort of 93 patients with COVID-19 followed by a test cohort of 40 patients, our predictive model produced an accuracy of 82.06% with an error of ±3 days in predicting the duration of viral shedding (Fig. 4H). A sparse model consisting of the top ten features was then validated using the validation set (30%, $n = 40$). The accuracy of using the top ten features was lower than that of using all features for viral shedding duration. The microbiome taxa that contributed the most to the model to determine the duration of viral shedding were from the three kingdom classes: *Adlercreutzia equolifaciens*, *Asaccharobacter celatus*, *Candida dubliniensis*, *Klebsiella phage vB KpnP SU5O*, and *Rhizobium phage vB RglS P1O6B* (Supplementary Fig. 6).

### Network analysis of the interactome of COVID-19 patients

We performed a network analysis of the interactions involving bacteriome, mycobiome, and virome to investigate the co-occurrence of multi-biome signatures in patients from the two clusters: Cluster 1 (severe) and Cluster 2 (non-severe). We first conducted a co-occurrence analysis by assessing the sparse compositional matrices approach to generate association networks. Taxa with close evolutionary relationships tended to be positively correlated, while distantly related microorganisms with functional similarities tended to be compete[25]. Herein, a positive interaction of microorganisms was defined by a correlative score representing the co-occurrence of microbes, while a negative value indicates co-exclusion (Sparcc |R| >0.1; $p < 0.05$). We found that patients in the non-severe cluster had a higher total number of bacteria and a lower number of viruses in the multi-interactome (Supplementary Fig. 7A). Intriguingly, we found an increased number of negative associations among bacteria, viruses, and fungi in the microbiome of a severe cluster (Supplementary Fig. 7A), suggesting a stronger co-occurrence of trans-kingdom patterns in patients with severe disease. We examined the network metrics of node degree, stress centrality, and betweenness centrality (of the nodes) to depict the impact of microbes on network integrity. The top representative taxa were not shared in the non-severe cluster. This observation suggests that the interactome of a microbe, rather than the microbe itself, dictates clinical status, such as the severity of COVID-19. We found more interactions involving bacteria-viruses and fungi-viruses in patients in the severe Cluster 1, including the invasive gut opportunistic pathogen *Ruminococcus gnavus*[26], fungi hubs of *Candida albicans*[27] and *Wickerhamomyces ciferrii* (reclassified and renamed *Pichia ciferrii*) (Supplementary Fig. 7C–E). In contrast, the core network in the non-severe cluster included more viruses, including *Bifidobacterium phage BigBern1*, *Streptococcus satellite phage Javan415*, and *Roseobacter phage DSS3P8* (Supplementary Fig. 7E). The results indicated clear segregation in terms of the patterns of nodes between the severe and non-severe cluster. Taking *R. gnavus* as an example, it was positively correlated with other constituent microbes in the severe cluster but negatively correlated in the non-severe cluster (Supplementary Fig. 7F). These findings highlight a preferential mechanism for the loss of inhibitory effect of pathogenic microbes in the severe cluster.

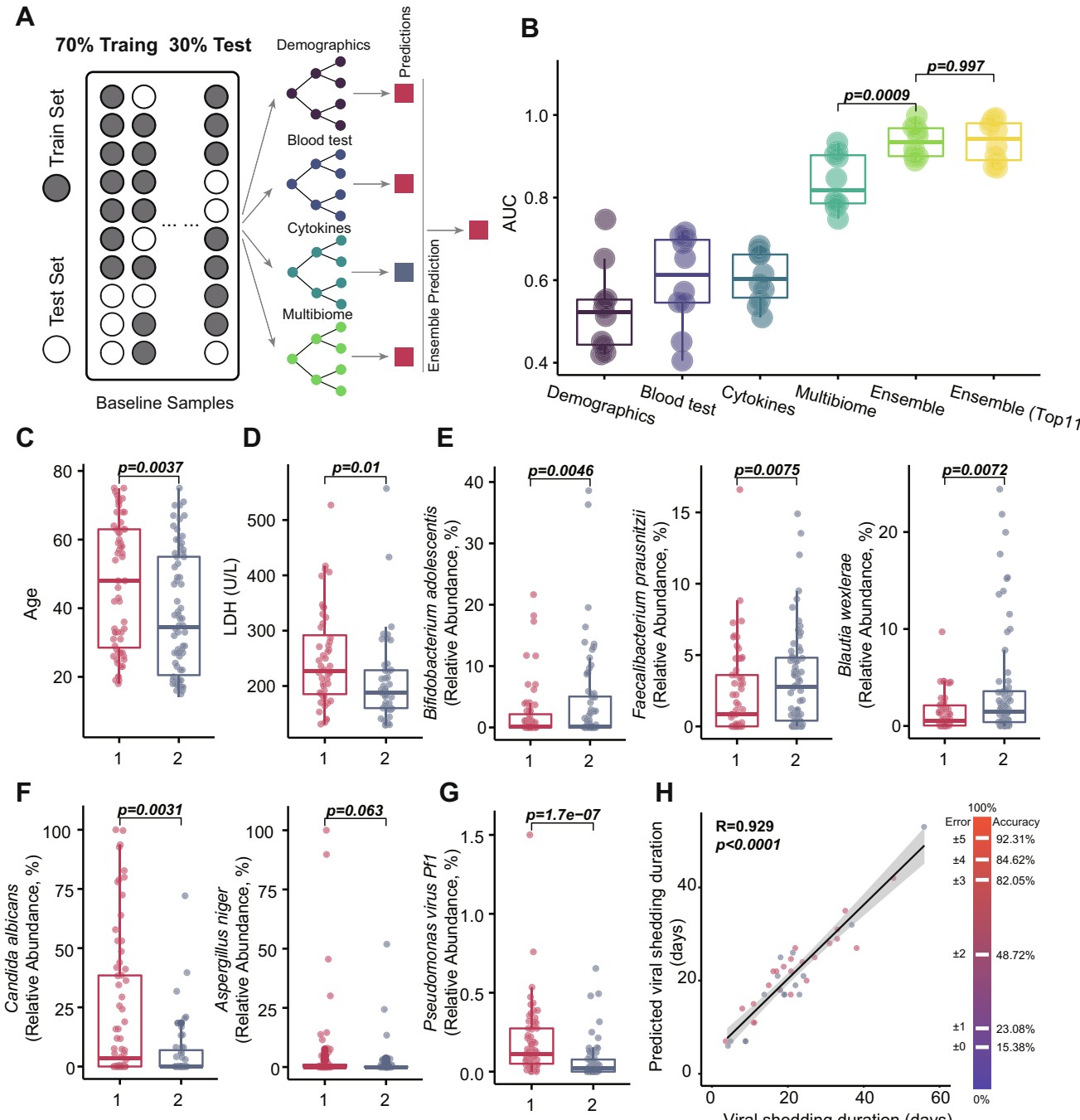

**Fig. 4 | Random forest classifier model trained on multi-biome and clinical data can predict the duration of viral shedding for individual COVID-19 patients.** **A** The input data is a vector with four components: demographics, blood tests, cytokines, and gut multi-biome profiles. To estimate model accuracies, a train-test sample split of 70% for training and 30% for testing was utilized. The testing data were then used to estimate the accuracy of the random forest model. **B** Box-and-whisker plot displaying the distribution of AUC scores for the cross-validation on the training set and the AUC scores for single measurements taken on the test set, obtained by random forest classification. Differences between groups were evaluated by the two-sided Wilcoxon rank-sum test. **C–G** Top features contribute to differentiating clusters (Cluster 1, n = 63; Cluster 2, n = 70) in the random forest

models. In **C–G**, the two-sided Wilcoxon test was used to check the differences between the two clusters. **H** Integration of multi-biome and clinical data for predicting the duration of viral shedding of SARS-CoV-2. The predicted positive time was paired with the real positive time for accuracy evaluation, and the accuracy was calculated at different error levels from ±0 to ±5 days. Error bands reflect the 95% CI. *R* and *P* values were calculated by two-sided Spearman Correlation, *p* < 0.0001. In **B–G**, the horizontal line in the boxplot indicates the median value. Box plots lower and upper hinges correspond to the first and third quartiles and upper and lower whiskers represent the highest and lowest values within 1.5 times the interquartile range.

## Discussion

Our cross-sectional and prospective multi-omics analyses reveal several new insights into the role of host and microbial factors in COVID-19 severity and long-term complications. First, we identified two

robust ecological clusters that defined severe COVID-19 and post-acute COVID-19. Second, these clusters, defined by altered multi-biome composition and impaired microbiome functionalities, were associated with PACS. Lastly, host and microbial factors can predict the

duration of respiratory viral shedding. Six host factors and five microbial candidates provided high accuracy, suggesting the prognostic potential of microbial markers for determining COVID-19 outcomes and consequences.

Several studies have demonstrated that the gut microbiota composition correlates with the severity of COVID-19 infection and persisted months after disease resolution[7]. The gut bacteriome has led to many discoveries of microbiota linked to disease progression in COVID-19[8], yet there is considerable untapped potential for non-bacterial microorganisms. Among the 133 patients, 110 were from the Prince of Wales Hospital, 17 from the United Christian Hospital, and 6 from Yan Chai Hospital. Since most (110/130) of the patients were assigned to the same hospital, which is nearest to their geographic location in Hong Kong, bias based on the geographic origins of patients should be limited in this study. There is considerable disease heterogeneity in COVID-19, given the variability in clinical, immunological inflammatory, and human fecal microbiome phenotypes. With the aid of data integration with a similarity network fusion approach for the multi-kingdom microbiome, we identified specific gut microbiome features that were linked to the severity, viral shedding duration, and post-acute complications of COVID-19. Evaluation of our model revealed that a combination of clinical information and gut microbiome data can substantially improve the differentiation capacities of the COVID-19 cohort. Among the microbiome and clinical variables, we found 11 factors, including bacteria, fungi, and viruses, which were significantly associated with cluster patterns and severe status. Using random forest modeling, we observed relationships between the features of the different multi-kingdom ecological constituents and the clinical features of patients with COVID-19. This embedding approach allowed us to connect these integrated multi-kingdom microbiome signatures to the specific clinically measurable features of the disease.

Multi-kingdom microbiota analyses provide new and previously unrecognized targets that could be considered as alternatives to, or used in combination with, established regimens for the prognosis of COVID-19. Particularly in the severe cluster, relationships with other kingdoms, such as fungi (*Candida glabrata, Candida albicans*) and viruses, are novel and previously unrecognized in COVID-19. The uncovered co-exclusion relationship between opportunistic pathogenic microorganisms and other species is particularly interesting, given the association between disease severity and long-term complications. The assessment of key influential taxa of microorganisms in different clusters highlights the relevance of integrative microbiome in the precision microbiome. The more severe cluster was associated with higher levels of *Candida albicans* and *Pseudomonas phages Pf1* and a lower abundance of *Bifidobacterium adolescentis*. The benefits of targeting influential microbes in an interactome, however, remain unknown and unaddressed in this work, and should be the focus of future studies.

Previous studies have reported that blood urea levels, an indication of kidney dysfunction, increase throughout infection[28]. Similarly, we found higher levels of urea in patients in the severe cluster than in those in the non-severe cluster. Moreover, functional microbiome analysis revealed that elevated urea might be explained by gut microbiome–mediated urea nitrogen recycling driven by *Klebsiella* species such as *K. pneumoniae* and *K. variicola*. Patients with severe COVID-19 exhibit abnormal bursts of the urea cycle in gut microbiome communities. We found that the involvement of gut microbes may hasten the accumulation of blood urea in COVID-19 patients. *Klebsiella* spp. are considered urease-producing and urea-hydrolyzing bacteria, which indicates that *Klebsiella* spp. can produce urease, an enzyme that catalyzes the hydrolysis of urea, to form ammonia and carbon dioxide[29]. Meanwhile, the enhancement of nitro-recycling may, in turn, cause an increase in serum urea, but the presence of impaired kidney function in COVID-19 patients may also need to be considered.

Eliminating pathogens to treat uremic toxins is a novel concept; however, if proven effective, it may have a significant impact on the management of patients with COVID-19.

Our study demonstrates an integrative microbiome approach; however, it has some limitations. First, the sample size was small, and our findings should be confirmed in larger cohorts across different populations. Besides, despite timely hospital admission and sample collection, there is also the possibility that patients were admitted at different stages of infection, which might be reflected in their viral load and gut microbiome composition. Despite the accelerated pace of advances in DNA sequencing and computational tools, bioinformatic techniques available for bacteriophage and phage crAss-like phages metagenomic libraries still have several inherent limitations. Reconstitution of the entire viral genome in the gut remains challenging. Future work and alternative approaches to the assessment of viromes, such as RNA sequencing, may yield different results and be more comprehensive, thereby enabling greater weighting of the vital contribution to the overall integrated microbiome, an important area of future exploration given the relatively poorly defined role of gut viruses in COVID-19. Bacteria, fungi, and viruses have been investigated; however, other types of microorganisms, such as archaea and protists, may also have important regulatory roles and require further exploration. Furthermore, although networks were weighted based on species richness and abundance, their true influence on the gut microbiome is not necessarily captured by richness and abundance alone, but rather by a function of functional genes, competition, substrate utilization, and energy flux through the ecosystem traits that cannot be comprehensively assessed by metagenomic sequencing alone. It is also important to test the robustness of the findings using publicly available subsets. An integrated modeling approach could be improved in the future with additional data concerning other immune markers, metabolomic data, and blood biomarkers.

Many emerging variants of COVID-19 continue to impose a global burden on healthcare systems. Ascertaining factors underlying differential susceptibility and poor outcomes following viral exposure is critical in improving public health responses and resource allocation via identification of those at high risk for severe disease and post-acute COVID-19 and their coordinated management through dedicated COVID-19 clinics. This study provides a compendium of gut multi-biome, immune response data, and an integrated framework to link gut microbiota to disease outcomes. By integrating patient microbiomes into either of the gut microbiome cluster identified in this study, we can begin to infer risk stratification and personalized management, and how microbiome therapeutic interventions may be most useful in specific patients. Our findings provoke the idea of future gut microbiome-based diagnostics and therapeutics based on an individual's multi-biome signature and propose applications of multi-omics technologies that could lead to an improved mechanistic understanding of microorganism–host interactions.

## Methods
### Study participants
Participants were recruited and consented under Research Ethics Committee (REC) no.

2020.076 and all subjects provided informed consent. This is a cross-sectional and prospective cohort study involving 133 patients with a confirmed diagnosis of COVID-19 (defined as a positive RT-PCR test for SARS-CoV-2 in the nasopharyngeal swab, deep throat saliva, sputum, or tracheal aspirate) hospitalized at three regional hospitals (110 from the Prince of Wales Hospital, 9 from the United Christian Hospital and 6 patients from Yan Chai Hospital) in Hong Kong, China between 13 March 2020 and 27 Jan 2021, followed-up to 6 months. Disease severity at admission was defined based on a clinical score of 1 to 5: (1) asymptomatic, individuals who tested positive for SARS-CoV-2 but who had no symptoms consistent with COVID-19. (2) mild,

individuals who had any signs of COVID-19 (e.g., fever, cough, sore throat, malaise, headache, and muscle pain) but no radiographic evidence of pneumonia; (3) moderate, if pneumonia was present along with fever and respiratory tract symptoms; (4) severe, if respiratory rate ≥30/min, oxygen saturation ≤93% when breathing ambient air, or PaO2/FiO2 ≤ 300 mm Hg (1 mm Hg = 0.133 kPa); or (5) critical, if there was respiratory failure requiring mechanical ventilation, shock, or organ failure requiring intensive care.[30] We defined post-acute COVID-19 syndrome (PACS) as at least one persistent symptom or long-term complication of SARS-CoV-2 infection beyond 4 weeks from the onset of symptoms which could not be explained by an alternative diagnosis. We assessed the persistence of the 30 most commonly reported post-COVID symptoms at 3 and 6 months after illness onset (Supplementary Data 8).

Patients who fulfilled the following criteria were eligible for analyses: (i) 18–70 years of age, (ii) no antibiotic therapy before at least 6 months, during, and 6 months after acute infection of SARS-CoV-2, (iii) no gastrointestinal symptoms during acute infection. Written informed consent was obtained from all patients. Dietary data were documented for all COVID-19 patients during the time of hospitalization (whereby standardized meals were provided by the hospital catering service of each hospital), and individuals with special eating habits, such as vegetarians, were excluded. After discharge, patients with COVID-19 were advised to continue a diverse and standard Chinese diet that was consistent with habitual daily diets consumed by Hong Kong Chinese. Data on the medical history, including age, gender, smoking status, and comorbidities (i.e., hypertension, diabetes mellitus, and hyperlipidemia), were recorded. Laboratory results include liver function tests (total bilirubin, creatine kinase, and LDH), renal function (urea and creatinine), complete blood count (i.e., hemoglobin, red blood cell, lymphocyte, monocyte, platelet, and polynuclear neutrophil), and CRP were collected.

## Stool samples
Stool samples were collected at admission from 133 patients and at 3 months and 6 months after discharge (average of three stool samples per subject). Stool samples from in-hospital patients were collected by hospital staff while discharged patients provided stools on the day of follow-up at 3 months and 6 months after discharge or self-sampled at home and had samples couriered to the hospital within 24 h of collection. Baseline (stools collected at admission) samples were the first sample after hospital admission and collected before antibiotic treatment. All samples were collected in tubes containing preservative media (cat. 63700, Norgen Biotek Corp, Ontario, Canada) and stored immediately at −80 °C until processing. We have previously shown that data on gut microbiota composition generated from stools collected using this preservative medium is comparable to data obtained from samples that are immediately stored at −80 °C[31]. The full sample list is summarized in Supplementary Data 9.

## Respiratory tract and stool SAR-CoV-2 viral load
Upper respiratory tract samples (pooled nasopharyngeal and throat swabs), lower respiratory tract samples (sputum and tracheal aspirate), and stool samples from 94 participants were collected at admission. We determined SARS-CoV-2 viral loads in these samples, using real-time reverse-transcriptase-polymerase chain-reaction (RT-PCR) assay with primers and probe targeting the N gene of SARS-CoV-2 designed by the US Centers for Disease Control and Prevention[32].

## Plasma cytokine measurements
Whole blood samples collected in anticoagulant-treated tubes were centrifuged at 2000×g for 10 min and the supernatant was collected. Concentrations of cytokines and chemokines were measured using the MILLIPLEX MAP Human Cytokine/Chemokine Magnetic Bead Panel− Immunology Multiplex Assay (Merck Millipore, Massachusetts, USA)

on a Bio-Plex 200 System (Bio-Rad Laboratories, California, USA). The concentration of N-terminal-pro-brain natriuretic peptide (NT-proBNP) was measured using Human NT-proBNP ELISA kits (Abcam, Cambridge, UK). Laboratory results at admission, including blood count test (platelet count, white blood cell count, neutrophil count) and the plasma concentrations of lactate dehydrogenase (LDH), C-reactive protein (CRP), albumin, hemoglobin, alkaline phosphatase, and aspartate aminotransferase, alanine aminotransferase, total bilirubin, and creatinine, were extracted from the electronic medical records in the Hong Kong Hospital Authority clinical management system.

## Quantification of fecal metabolites
The quantification of fecal metabolites from 79 fecal samples at admission was performed by Metware Biotechnology Co., Ltd. (Wuhan, China). Acetic was detected by GC-MS/MS analysis. Agilent 7890B gas chromatography coupled to a 7000D mass spectrometer with a DB-5MS column (30 m length × 0.25 mm i.d. × 0.25 μm film thickness, J&W Scientific, USA) was used. Helium was used as a carrier gas, at a flow rate of 1.2 mL/min. Injections were made in the splitless mode and the injection volume was 2 μL. The oven temperature was held at 90 °C for 1 min, raised to 100 °C at a rate of 25 °C/min, raised to 150 °C at a rate of 20 °C/min, and held at 150 °C for 0.6 min, further raised to 200 °C at a rate of 25 °C/min, held at 200 °C 0.5 min. After running for 3 min, all samples were analyzed in multiple reaction monitoring modes. The temperature of the injector inlet and transfer line were held at 200 and 230 °C, respectively. L-isoleucine and L-arginine were detected by LC-MS analysis. LC-ESI-MS/MS system (UPLC, ExionLC AD, https://sciex.com.cn/; MS, QTRAP® 6500+ System, https://sciex.com/) was used for analysis. The analytical conditions were as follows, HPLC: column, Waters ACQUITY UPLC HSS T3 C18 (100 mm × 2.1 mm i.d.,1.8 μm); solvent system, water with 0.05% formic acid (A), acetonitrile with 0.05% formic acid (B). The gradient was started at 5% B (0–10 min), increased to 95% B (10–11 min), and ramped back to 5% B (11–14 min); flow rate, 0.35 mL/min; temperature, 40 °C; injection volume: 2 μL. The ESI source operation parameters were as follows: an ion source, turbo spray; source temperature 550 °C; ion spray voltage (IS) 5500 V (Positive), −4500 V (Negative); DP and CE for individual MRM transitions were done with further DP and CE optimization.

**Stool DNA extraction and sequencing**. Detailed methods for extracting bacterial and fungal DNA are described in ref. 8. Briefly, the fecal pellet was added to 1 mL of CTAB buffer and vortexed for 30 seconds, then the sample was heated at 95 °C for 5 min. After that, the samples were vortexed thoroughly with beads at maximum speed for 15 min. Then, 40 μL of proteinase K and 20 μL of RNase A were added to the sample and the mixture was incubated at 70 °C for 10 min. The supernatant was then obtained by centrifuging at 13,000×g for 5 min and was added to the automated Maxwell RSC machine (Promega, Wisconsin, USA) for DNA extraction. The total viral DNA was extracted from each fecal sample, using TaKaRa MiniBEST Viral RNA/DNA Extraction Kit (Takara, Japan) following the manufacturer's instructions. Extracted total viral DNA was then purified by the DNA Clean & Concentrator Kits (Zymo Research, CA, USA). After the quality control procedures by Qubit 2.0, agarose gel electrophoresis, and Agilent 2100, extracted DNA was subject to DNA libraries construction, completed through the processes of end repairing, adding A to tails, purification, and PCR amplification, using Nextera DNA Flex Library Preparation kit (Illumina, San Diego, CA). Libraries were subsequently sequenced on our in-house sequencer Illumina NextSeq 550 (150 base pairs paired-end) at the Center for Microbiota Research, The Chinese University of Hong Kong. Raw sequence data generated for this study are available in the Sequence Read Archive under BioProject accession: PRJNA714459.

**Bioinformatics.** Raw sequence data were quality filtered using Trimmomatic V.39 to remove the adapter, low-quality sequences (quality score <20), and reads shorter than 50 base pairs. Contaminating human reads were filtering using Kneaddata (V.0.7.2 https://bitbucket.org/biobakery/kneaddata/wiki/Home, Reference database: GRCh38 p12) with default parameters. Following this, microbiota composition profiles (bacteria and fungi) were inferred from quality-filtered forward reads using MetaPhlAn3 version 3.0.5[33] and MiCoP[34]. Micop has been proven to be more effective for eukaryotes identification in human microbiome data[34]. GNU parallel[35] was used for parallel analysis jobs to accelerate data processing.

**Viral profiling of metagenomics data.** Identification of viral sequences in the process of viral metagenomic analysis is notoriously challenging due to the lack of a universal viral marker as opposed to bacterial 16 S rRNA, for example. Thus, reference-based read mapping is limited by a scarcity of annotated viral genomes. We used an optimized pipeline, capable of de novo extraction and retrieval of viral contigs from shotgun sequencing reads. Raw sequence quality was assessed using FASTQC and filtered utilizing Trimmomatic using the following parameters; SLIDINGWINDOW: 4:20, MINLEN: 60 HEADCROP 15; CROP 225. Contaminating human reads were filtering using Kneaddata (Reference database: GRCh38 p12) with default parameters. Megahit[36], with default parameters, was chosen to assemble the reads into contigs per sample. Assemblies were subsequently pooled and retained if longer than 1 kb. Bacterial contamination was removed by using an extensive set of inclusion criteria to select viral sequences only. Briefly, contigs were required to fulfill one of the following criteria; 1) Categories 1–6 from VirSorter when run with default parameters and Refseqdb (−db) (1)[37] positive, (2) circular, (3) greater than 3 kb with no BLASTn alignments to the NT database (January '19) (e-value threshold: 1e-10), (4) a minimum of 2 pVogs with at least 3 per 1 kb[38], (5) BLASTn alignments to viral RefSeq database (v.89) (e-value threshold: 1e-10), and (6) less than three ribosomal proteins as predicted using the COG database[39]. HMMscan was used to search the pVOGs hmm profile database using predicted protein sequences on VLS with an e-value filter of 1e-5, retaining the top hit in each case. Afterward, a fasta file combining viral contigs was compiled. The redundant sequences were eliminated by CD-HIT-EST provided from CD-HIT 4.8.1. This viral database includes the viral contigs recovered by the screening criteria from the bulk metagenomic assemblies. Then the paired reads were mapped to the viral contig database with BWA, using default parameters. The viral operational taxonomic unit (OTU) table of viral abundance was pulled from BWA sam output files by script, and normalized by the number of metagenomic reads and the OUT sequence length. The contigs were analyzed according to their open reading frames (ORFs). The ORFs on the contigs were predicted using MetaProdigal (Hyatt et al., 2012) (v2.6.3) with the metagenomics procedure (-p meta). To annotate the predicted ORFs, the amino acid sequences of the ORFs were queried by Diamond[40] against the viral RefSeq protein (v84) with an E-value <$10^{-5}$ and a bitscore >50. The viral Refseq proteins with the top closest homologies (E-value <$10^{-5}$ and bitscore >50) were considered for each ORF, analogous to a previously reported method[41].

**Integration and clustering analysis of multi-biome data**
For each biome dataset, microbes prevalent in at least 5% of patients (that is, $n \geq 7$) with an average abundance of 1% were kept for analysis (Detected 737, Kept 242, Removed 495). Integration of bacterial, fungal, and viral community data was performed by weighted SNF (WSNF) using an online tool (https://integrative-microbiomics.ntu.edu.sg)[14]. Briefly, the respective weights of each biome are assigned based on the richness of the data, as demonstrated by the number of species present in each biome. Using the merged dataset (bacteria, fungi, and viruses), the tool generates a corresponding patient similarity network

using a spectral clustering algorithm with the default settings (Bray–Curtis), outputting the cluster assignments for each patient. The optimal number of clusters ($n = 2$) was determined by WSNF using the eigengap method and the value of K nearest neighbors, which was set based on the optimal silhouette width[14].

**Random Forest stratification.** R package random Forest v4.6–14 was used to develop a stratification model of patients in different clusters. Four datasets from 133 patients, including demographic, blood tests, cytokines, and multibiome were used separately or in combination (ensemble) to train the model for cluster stratification. Machine learning models were first trained on the training set (70%, $n = 93$) with fivefold cross-validation, and then were applied to the test set (30%, $n = 40$) for validation. Each time a new feature was added to the model, the performance of the model was re-evaluated using the above training and validation set. This process was repeated ten times to obtain a distribution of random forest prediction evaluations. The training dataset (70%) was used for feature selection. A trained forest produces a variable importance list based on a mean decrease in the Gini index. The feature importance vector (mean decrease Gini index), including weights for every species, demographic, blood test, or cytokines predictive capacity was collected. The final model for stratification was chosen when the best overall AUC value was achieved. For the construction of an optimal prediction model in the ensembled dataset, the importance value of each feature to the stratification model was evaluated by recursive feature elimination first, and then the selected features are added to the model one by one according to the descending importance value. The hyperparameters for the random forest model were ntree = 10,000, Gini index as impurity criterion, and the default square root of the number of features (species in this case) to use for each split in the decision tree.

**Random Forest regression analysis for positive time prediction.** The random forest regression model was used to regress features from ensembled dataset (demographic, blood test, cytokines, and multibiome) in the time-series profiling of COVID-19 patients against their SARS-CoV-2019 positive time (Upper respiratory tract) using default parameters of R package randomForest v4.6–14 (ntree = 10,000, using default mtry). The dataset was divided into 70% training and 30% testing set. The RF algorithm, due to its non-parametric assumptions, was applied and used to detect both linear and nonlinear relationships between multiple types of features and positive time[42], thereby identifying features that discriminate different viral persistent duration in COVID-19 patients. The top-ranking important positive duration-discriminatory features required for prediction were identified based on "rfcv" function in the randomForest package. Ranked lists of important features in order of reported feature importance were determined over ten times fivefold of the algorithm on the training set (70%, $n = 93$). Using the profiles of a multi-microbiome, demographic, blood test, and cytokines, the performance of models was further evaluated with a fivefold cross-validation and repeated ten times to obtain a distribution of random forests prediction evaluations. The final model for regression was chosen when the best overall accuracy was achieved. The predicted positive time was paired with the real positive time for accuracy evaluation, and the accuracy was calculated at different error levels from ±0 to ±5 days.

**Co-occurrence analysis of microbial interaction within COVID-19 patient clusters**
SparCC[43] was used to identify co-occurrence correlations among bacteria, fungi, and virus from the R package "SpiecEasi v1.1.1" with 20 iterations in the outer loop and 10 iterations in the inner loop[44]. The correlation strength exclusion threshold was 0.1 using the SparCC default setting. Absolute values of correlations below 0.1 are considered zero by the inner SparCC loop, and $p$ value below 0.05 was

considered significant. The resulting network was characterized and visualized via Cytoscape (v3.9.1).

**Statistical analysis and inferring gut microbiota composition.** Continuous variables of demographic features were expressed in the median (interquartile range), whereas categorical variables (disease severity, five-point scale of severity) were presented as numbers. Qualitative and quantitative differences between subgroups were analyzed using chi-squared or Fisher's exact tests for categorical parameters and the Wilcoxon test for continuous parameters, as appropriate. The odds ratio and adjusted odds ratio (aOR) with a 95% confidence interval (CI) were estimated using logistic regression to examine clinical parameters associated with the development of PACS. The site-by-species counts and relative abundance tables were input into R V.3.5.1 for statistical analysis. Principal Coordinates Analysis (PCoA) was used to visualize the clustering of samples based on their species-level compositional profiles. Associations between gut community composition and patients' parameters were assessed using permutational multivariate analysis of variance (PERMANOVA). Associations of specific microbial species with patient parameters were identified using the linear discriminant analysis effect size (LEfSe) and the multivariate analysis by linear models (MaAsLin2) statistical frameworks implemented in the Huttenhower Lab Galaxy instance (http://huttenhower.sph.harvard.edu/galaxy/). PCoA, PERMANOVA, and Procrustes analysis are implemented in the vegan R package V.2.5–7.

### Reporting summary

Further information on research design is available in the Nature Portfolio Reporting Summary linked to this article.

## Data availability

The raw sequences generated in this study have been deposited in the NCBI Sequence Read Archive (SRA) database under accession PRJNA876804. MS metabolomics data have been deposited in the EMBL-EBI MetaboLights database with the identifier MTBLS6317. Source data are provided with this paper.

## Code availability

All bioinformatic and machine learning model scripts are available on Github [https://github.com/g-micro/multi-omics][45].

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

## Acknowledgements

This work was supported by InnoHK, The Government of Hong Kong, Special Administrative Region of the People's Republic of China. We would like to thank all healthcare workers working in isolation wards of the Prince of Wales Hospital, United Christian Hospital, and Yan Chai Hospital, Hong Kong SAR, China. We thank Joey Chan, Dai Min, Lok Cheung Chu, and other staff/students for their technical contribution to this study, including sample collection, inventory, and processing, and Hui Zhan for assistance with DNA extraction and sequencing. Thanks to Gabriel Lee for proofreading the article.

## Author contributions

Q.L. and Q.S. conceived the study, developed algorithms, ran analyses, and took responsibility for the integrity of the data and preparation of the manuscript. F.Z. contributed to part of the metabolites analysis. H.M.T. provided critical comments on the manuscript. J.W.Y.M., G.C.-Y.L., S.S.S.N., J.YL.C., A.L., and C.P.C. contributed to participant recruitment, sample collection, and biobank management. C.L. and W.L. contributed to metagenomic sequencing. D.SC.H., P.KS.C., and F.K.L.C. contributed to the study design and data interpretation. SCN contributed to the study design, data analysis, and manuscript writing. All authors gave final approval for the version to be published. All authors agree to be accountable for all aspects of the work in ensuring that questions related to the accuracy or integrity of any part of the work are appropriately investigated and resolved.

## Competing interests

F.K.L.C. and S.C.N. are the scientific co-founders and sit on the board of Directors of GenieBiome Ltd. S.C.N. has served as an advisory board member for Pfizer, Ferring, Janssen, and Abbvie and is a speaker for Ferring, Tillotts, Menarini, Janssen, Abbvie, and Takeda. She has received research grants from Olympus, Ferring, and Abbvie. F.K.L.C. has served as an advisor and lecture speaker for Eisai Co. Ltd., AstraZeneca, Pfizer Inc., Takeda Pharmaceutical Co., and Takeda (China) Holdings Co. Ltd. S.C.N., K.L.F.C., and Q.L. are inventors of a patent application (US provisional patent application no. 63/355,443) in connection with this work. The remaining authors declare no competing interests.
