## [Peer Review File · Nature Communications]

REVIEWER COMMENTS

Reviewer #1 (Remarks to the Author):

Qin Liu et al. present an analysis of many types of microbiome and host data that effectively predict patient COVID-19 outcomes. As far as I can tell, the study was done well and the analyses sound.

I have no major complaints.

Intermediate concerns:

1. Overall, the grammar is not horrible, but has room for improvement, particularly in the discussion.

Here are some examples:

Line 39: odd phrasing to say "viral load of respiratory samples"

Line 90: This sentence is confusing and seems like cytokines are part of viral RNA load quantification? Rephrase or perhaps break into two sentences.

Line 93: Rephrase. Fecal samples don't have time points -- or at least that's not what was done. This sounds like one collected fecal sample was processed at three time points. Also, since not all patients had three fecal samples, I would recommend saying "up to three longitudinal time-points."

Line 133: change "function" to "functions"

Line 259/260: As written this is hard to understand: "Our evaluation on model revealed that including clinical information in addition to gut microbiome..."

Line 290: This is phrased oddly. Is it being suggested that Klebsiella is a (beneficial) symbiont?

Line 315: "By indicating patients' microbiomes into either..." Perhaps the word 'integrating' was intended?

2. Network analyses: Are all the discussed and displayed associations statistically supported? Perhaps it is my ignorance of these type of analyses, but without seeing statistics, these results seems like speculation. I think that is fine to some extent, but as such, Figures 5 and 6 may be better suited for Supplementary Figures. These figures also don't seem particularly informative.

Minor points:

1. Other microbes. Since an effort was made to consider fungi and virus, I wonder if other components of the microbiome were considered? -- some archaea should meet the 5% threshold, and depending on the population, some protists like Blastocystis.

Not that they would necessarily be important, but they could be.

2. Please be more consistent with style. For example:

Line 42: Cluster 1 vs cluster 1 (capitalization or not)

Line 179: Pseudomonas phages Pf1 vs. Line 184: Pseudomonas virus Pf1

Figures: All p-values should be written consistently, probably with "p =" (not just a number), with same font size (e.g., Supp Fig 1), same bolding and italics or not (e.g., Supp Fig 4), same upper or lower case (e.g., Fig 3 legend), and the same format for exponentials (e.g., Supp Fig 3).

In addition, many times the taxa are written with "_" instead of spaces. Please remove underscores.

3. Pathogens.

Line 46, 164, 215, 272: Are these really pathogens? I'm guessing maybe they've been shown to be opportunistic pathogens? If so, be sure to use the word opportunistic and cite where they've been shown to be pathogenic.

Specific line comments:

Line 46: Aspergillus and C. albicans are spelled incorrectly here and some other places.

Line 52 and 103: Is it important to show significant digits to the hundredth place?

Line 72: The notion of fungal communities in the healthy human GI tract is controversial, as most fungi tend to be dead and only transiently present from saliva and consumed foods.

I won't make a fuss about using the possibly technically incorrect term "mycobiome", but here perhaps just remove the word "communities" and say ... of viruses and fungi which...

Line 103: WSNF was defined, but not WSF.

Line 109: MaAslin. I believe the L is also capitalized.

Line 119: CXCL10 has not been mentioned previously. What is it?

Line 146: ALP and ALT have not been defined previously.

Line 165: Why were the 2nd and 3rd most discriminant bacteria specifically mentioned? (ignoring #1)

Line 172: Which time points are considered for this analysis? All?

Any thoughts on differences between Fig 4e-g and the taxa identified by MaAsLin?

Line 181: forest not forrest

Line 181: Why were 11 factors chosen?

Did this perform better than the top 10, top 12, or other numbers?

Line 197: What value says what taxa contributed most to the model? Is it possible to add that information? Looking at Supp Fig 6, there is no indication that fungi and viruses contributed more than bacteria. Perhaps that value would also explain why the three specific taxa are pointed out on line 198/199. Also, the last part of this sentence does not make sense. It's odd to say these taxa could be considered after they were just considered.

Line 248: Remove "For example,"

Line 349: One could make the case that most people have special eating habits. I'm surprised to see vegetarians excluded. What fraction of individuals were excluded due to dietary preferences?

Line 406: How deep was the sequencing? If it's stated, I don't see it.

Line 412: Maxwell RSC machine (what company?)

Line 415: I'm guessing the phrase "to obtain viral DNA, respectively" is not needed (what else would viral DNA kits be used for?)

Line 430: I'm familiar with Metaphlan, but have never heard of MiCoP. This should either be explained or referenced.

Line 680/681: What is the y-axis for (C)?

Specific figure comments:

Figure 2: (A) Adjust position of months, (C) Bifidobacterium (spelling)

Here and elsewhere I find the color schemes confusing. It is fine that red is Cluster 1 throughout and blue is Cluster 2 throughout, but then it is confusing that bacteria are colored the same red, the mycobiome is colored the same blue, and the virome is colored the same green as the chi squared tests.

Figure 3: Fix spelling for fatigue, dizziness, and align taxa names better with bars

Figure 4:

4B +C axis titles are bold and the others are not. Taxa names should be italicized.

4H. Needs more information than Positive Time. Not clear until read text or legend that referring to viral shedding.

Figure 5D: If the large central blue dot is *C. albicans*, where is the dot for *Wickerhamomyces*?

Supp Fig 1: Definitions are needed of what the boxplots mean (line = median, etc).

The stats for B. should be moved from A to B.

Define the circles in B (centroids?)

I don't see any mention of Supp Fig 2C in the text.

Supp Fig 3D: I doubt the p-value is truly 0. Perhaps put <0.01 ?

Supp Fig 5C+D: I was very confused why there is essentially no difference between Cluster 1 and 2. But now I think it is just a color scheme issue, as the legend says baseline (red like Cluster 1) and 6-month follow-up (blue like Cluster 2). Also, reading the legend I see that C = Cluster 1 and D = Cluster 2. This should be labeled on the figure, just like with A + B.

Reviewer #2 (Remarks to the Author):

Summary

Liu et al. present in their manuscript an analysis of COVID-19 patients with and without post-acute COVID-19 syndrome based on their gut microbiome profiles, and additional demographic and clinical data. While the herein included data and the described analysis are interesting and valuable, not least in the light of the ongoing pandemic, the manuscript often lacks important information as well as a more thorough discussion. In particular, the code and data required to reproduce the study are missing, and many passages in the Results and Methods sections are confusing (e.g. inconsistency in sample numbers and insufficient description of the analysis). Most importantly, the authors have already published a very similar study, which is also being cited in the present manuscript (<http://dx.doi.org/10.1136/gutjnl-2021-325989>, reference 10). However, the newly obtained results described here are not being discussed in the context of the previously published study.

Major comments

Comment 01

The authors demonstrate interesting correlations of the taxa with COVID-19 severity. However, aside from the single mention of the use of multi-omics for weighted similarity networks, the use of multi-omics is very limited. Therefore the title could be reassessed and revised. Importantly, given the

availability of other COVID-19 datasets, it may be important to validate their in silico findings in a random publicly-available subset to ensure their robustness.

Comment 02

The post-acute COVID-19 syndrome (PACS) is defined as the presence of long-term complications observed after 4 weeks after the onset of symptoms (L338-340). This is a rather short time period and it is not clear from the manuscript whether these symptoms were reassessed at the 3 and 6 months time points to reevaluate their persistence.

Comment 03

Reproducibility.

The authors do not provide the information required to reproduce their study: there is no reference to a code repository to reproduce the described analysis; there is no raw data accession in the Data Availability statement and the one provided in the Methods section (i.e., PRJNA714459) is from another already published study; there is no sample metadata including the demographic information, laboratory measurements, PACS symptoms etc.; there is no processed data such as the microbiome profiles, built prediction models, and data used to generate the figures.

Comment 04

Sample numbers.

There seems to be some discrepancy in the number of samples and missing information on the sample size for some analyses. For example, the Abstract states 296 fecal metagenomes, however 293 is mentioned in L92. Furthermore, the authors describe the assessment of only 79 fecal samples for metabolomics (L94). It is unclear why or how this number was arrived upon. Additionally, it is not evident whether the profiles from 79 samples are representative of the 133 patients (or 296 fecal) samples collected. Similarly, the viral load was assessed only for 94 samples (upper and lower respiratory tract, and fecal samples, L373). There is also no information on the missing stool samples: there have to be $3 \times 133 = 399$ fecal samples in total so there are 103 to 106 missing samples. It is not stated whether the missing samples affect certain time points more than others.

A full list of the samples with the appropriate metadata should be provided for review. Any sample that was not used for each of the various assays should be clearly indicated with justification. In addition, statistical justification along with a stratification of the different patient/sample groups is required.

Comment 05

Found patient clusters and comparison to previous studies.

It is interesting to see two distinct clusters based on the similarity network approach. However, the description lacks some details. For example, it is not clear whether “pooling” in L101 is referring to the weighted similarity network fusion approach and whether “baseline” in L97 describes the time point “at admission”.

The authors state that the found two clusters had similar demographic parameters except for age: Cluster 1 contained older patients than Cluster 2 (L122-123). Given that the risk of higher disease severity for COVID-19 increases with age, it cannot be ruled out that the found clusters represent different age groups and that the observed differences in microbiome profiles and symptoms might be the consequence of that. The authors should consider this option and investigate whether patients’ age is correlated with their symptoms.

For the time point “at admission”, there is also the possibility that the patients were admitted to the hospital at different stages of infection which might be also reflected in their microbiome and viral load. The authors should discuss this in the manuscript as it might also have contributed to the observed cluster formation.

Could there be a bias based on the geographic origins of the patients? It is also likely that the “severe patients” were housed in the same if not similar wards, compared to those with milder symptoms. Based on the Methods section, the possible biases induced due to housing conditions at the hospital (i.e. rooms) are unaccounted for. A reader would benefit from such information which can be contributing factors to microbiome profiles (see <https://doi.org/10.1128/msphere.01007-21>).

Finally, the authors state that the microbiome was stable over time within the clusters (L159-161). Furthermore, the majority of patients in Cluster 1 (84%) had PACS while only 44% were present in Cluster 2 (Fig. 3C, L166-168). However, in their other study, the authors also describe a recovery of the gut microbiome in patients without PACS at 6 months after disease resolution (<http://dx.doi.org/10.1136/gutjnl-2021-325989>, reference 10). Based on that, one would expect to see at least some temporal changes in Cluster 2 containing patients who did not develop PACS. Given this discrepancy, the authors should additionally compare the microbiome profiles (alpha- and beta-diversity) of patients with and without PACS and also across the different time points. In general, while the authors do cite their previous study in the present manuscript they do not compare the newly obtained results to previous observations made in COVID-19 patients, though both studies have certain similarities in their design.

Comment 06

Microbiome profiling and data integration.

The authors state that the microbiome profiling was done to generate datasets representing three biomes: viruses (using a custom assembly-based approach), bacteria and fungi (using MetaPhlan3 and MiCoP). These data sets were then combined using weighted similarity network fusion (WSNF). There are multiple points which have to be clarified regarding this part of the analysis.

1) MetaPhlan3 includes in its database information bacterial, archaeal, viral, and eukaryotic genomes. Similarly, MiCoP estimates abundance of viral and fungal organisms. It has to be clarified whether the authors used all obtained hits from these two tools or if they limited the output only to specific organism groups, e.g. bacteria and fungi. If all hits were retained then it should be stated how the authors deduplicated the results to avoid multiple hits to the same species from different tools and approaches.

2) The WSNF method assigns weights to the different biomes based on their richness: the number of species found in each biome. The number of observed species is highly affected by the used tools and their databases. Does the used approach account for that? Moreover, the authors should reference the respective publication describing the used WSNF approach.

3) The authors should state why they decided to pursue the assembly-based custom approach for virome profiling instead of using the output from MetaPhlan3 and/or MiCoP.

4) It is not completely clear from the text whether the integration of the three biome datasets was done using all available samples or for each time point separately (i.e., at admission, 3 months, and 6 months).

Comment 07

L99/L457-458: The authors conveniently use a prevalence cutoff of 5% when filtering the dataset. However, it is unclear how they arrive at this number. Additionally it should be justified with appropriate references (see PMID 33510727). Also, it is not clear how the “average abundance of 1%” was used for filtering and why the authors chose the 1% cutoff. Moreover, the authors should state how many features were removed from the dataset using their filtering criteria and how many remained and were used for further analysis.

Comment 08

L137: The authors claim that blood urea levels were correlated with disease severity. However, based on the figures, a majority of the points are beyond the 2x standard deviation, suggesting a potential artifact due to outlier samples. This analysis has to be adjusted for outlier effects and assessed critically.

Comment 09

L205: The authors claim an ‘ensemble-based’ method for the network analyses. However, the Methods section only describes using the “Reboot” approach. It must be noted that Reboot is quite outdated (2012) and since then additional/better methods for assessing sparse compositional matrices have been introduced. This is evident from Schwager et al.’s work (PMID: 29140991). It may be prudent to use an alternative method to corroborate the findings described herein.

Comment 10

L218: The authors demonstrate an example with *C. spiroforme*. However, it is unclear why this taxon was chosen. Especially given the fact that *R. gnavus* was also part of the description. It may be more prudent to use *R. gnavus* as an example to state the case, or explain why it did not show the expected correlations with the severe cluster.

Comment 11

L226: It is surprising to observe that the “density” of the network is low, and yet the connected component value is 1. Could the authors explain the phenomenon driving this? In network theory, a highly connected component usually arises from a highly dense network, so this finding is counter-intuitive.

Comment 12

For the patient stratification model, it seems that the clustering results obtained from the complete dataset were used as patient labels. If this is the case, then there might be an overfitting issue as the clusters and their optimal number were defined based on all available data. Therefore, the validation set used later does not completely represent truly unseen data as described by the authors (L472). A cleaner approach would be to perform the clustering using only the microbiome profiles of the training set. Expected labels for the validation set could be generated, for example, by a k-nearest neighbors approach to be able to compute the model accuracy. Another point is the feature ranking and selection procedure. Here it is not clear how exactly this was done (i.e., which data was used for feature selection, training and validation) and what the final model is. Additionally, the authors should not only look at accuracy but also consider other performance measures and their variance (in addition to the mean value).

Similarly, the paragraph about the RF model used to predict the time period of patients being SARS-CoV-2 positive lacks some details to understand the performed model training and validation procedure. The authors should try to provide a more concise and clear description of this analysis step.

Minor comments

L204-205: It would be better to not introduce additional names or labels for Cluster 1 and Cluster 2.

L46: *R.gnavus* are commensal taxa found in many mammalian microbiomes. They are not pathogenic bacteria, unless in a particular context such as septic arthritis. Please refrain from using this misleading description

L90-92: The text reads such that viral loads were checked in PBMCs. Please rephrase.

L111-114: Please show the 95% and 99% confidence intervals and provide the PERMANOVA analyses results as a supplementary table with special emphasis on the F statistic.

L130: Additional details regarding how the functional profiling was performed are needed.

L138: Additional details about how specific microbiome species were associated with elevated urea levels are needed.

L209-211: Please provide the correlation coefficient and p-value cutoffs in the results with respect to what is termed a “co-occurrence” and what is not.

L344: It is unclear if all the patients were housed in the same hospital/quarantine facility. Please elaborate.

L363-364: Why do the authors mention the antibiotic treatment if having no antibiotics therapy 6 months before, during and 6 months after the SARS-CoV-2 infection was one of the inclusion criteria (L344-346)?

L379: Were the whole blood samples collected at admission?

L386-404: Which samples were analyzed here? All 293 stool samples?

L432: The subsection title “Bioinformatic Viral Processing” is confusing. E.g., “Viral profiling of metagenomics data” or similar would be more appropriate.

L437: Were the contigs deduplicated?

L448-450: How were the OTUs defined and generated from the reads mapped to contigs? Why was the normalization done by total number only and not also by the OTU sequence length?

L464-465: Which software was used for patient clustering? What is the relationship between the eigengap method to choose the optimal number of clusters and the number of nearest neighbors?

L467: While the features used for model training are described in this paragraph the authors do not explicitly say what the response variable is.

L470-473: Although the training and validation datasets are described, the ‘test’ dataset is missing in the description. Please clarify.

L489-493: Is the result of the training procedure the “sparse model” or how exactly was that model generated after ranking the features?

L503-504: Which variables are described here? Why were categorical variables represented as numeric values (numbers and/or percentages)? Was the microbiome data normalized or transformed?

L538: USPTO application number is missing.

REVIEWER COMMENTS

Reviewer #1 (Remarks to the Author):

Qin Liu et al. present an analysis of many types of microbiome and host data that effectively predict patient COVID-19 outcomes. As far as I can tell, the study was done well and the analyses sound. I have no major complaints.

Response: We thank the reviewer for the positive comments. We present a revised version of the manuscript based on this reviewer's additional comments.

Intermediate concerns:

1. Overall, the grammar is not horrible, but has room for improvement, particularly in the discussion. Here are some examples:

Following the Reviewer's comment and advice we have now
Line 39: odd phrasing to say "viral load of respiratory samples"

Response: Thank you. We have rephrased this sentence to "viral load in 1,378 respiratory tract samples". (Line 40)

Line 90: This sentence is confusing and seems like cytokines are part of viral RNA load quantification? Rephrase or perhaps break into two sentences.

Response: We have divided this sentence into two sentences. (Line 92-95)

"We assessed viral RNA level in nasopharyngeal swabs and fecal samples using quantitative PCR with reverse transcription (RT-qPCR). We also assessed plasma cytokines and chemokines levels and leukocyte profiles in freshly isolated peripheral blood mononuclear cells (PBMCs) using flow cytometry."
"

Line 93: Rephrase. Fecal samples don't have time points -- or at least that's not what was done. This sounds like one collected fecal sample was processed at three time points. Also, since not all patients had three fecal samples, I would recommend saying "up to three longitudinal time-points."

Response: We have rephrased the sentence to " We also analyzed gut microbiome composition (bacteria, virus, fungi) in 296 serial faecal samples collected at up to three longitudinal time-points from admission to six months after virus clearance..." (Line 95-97)

Line 133: change "function" to "functions"

Response: We have corrected "function" to "functions". (Line 141)

Line 259/260: As written this is hard to understand: "Our evaluation on model revealed that including clinical information in addition to gut microbiome..."

Response: We have rephrased this sentence to "Evaluation of our model revealed that a combination of clinical information and gut microbiome data can achieve a substantial improvement in..." . (Line 268-269)

Line 290: This is phrased oddly. Is it being suggested that Klebsiella is a (beneficial) symbiont?

Response: We thank the reviewer for pointing out this ambiguity. We have rephrased this sentence: "Eliminating pathogens to treat uremic toxins is a novel concept..."(Line 299-300)

Line 315: "By indicating patients' microbiomes into either..." Perhaps the word 'integrating' was intended?

Response: The word "indicating" is replaced by "integrating". Line 329

2. Network analyses: Are all the discussed and displayed associations statistically supported? Perhaps it is my ignorance of these type of analyses, but without seeing statistics, these results seems like speculation. I think that is fine to some extent, but as such, Figures 5 and 6 may be better suited for Supplementary Figures. These figures also don't seem particularly informative.

Response: Thank you for the suggestion. We used the SparCC approach to analyse the association, and a p value below 0.05 was considered to be significant. As per the reviewer's suggestion, we have moved the figures to supplementary Figure 7. We have also shortened this section in the manuscript at Line 222-248.

Minor points:

1. Other microbes. Since an effort was made to consider fungi and virus, I wonder if other components of the microbiome were considered? -- some archaea should meet the 5% threshold, and depending on the population, some protists like Blastocystis. Not that they would necessarily be important, but they could be.

Response: Thank you for the suggestion. We analysed archaea in the samples and only one species of *Methanosphaera stadtmanae* was detected . The prevalence was 4.2%, which is lower than 5% threshold. We have included a sentence in the revised paper "Bacteria, fungi, and viruses were investigated but other types of microorganisms, such as archaea and protists, may also have important regulatory roles and require further exploration." (Line 314-316)

2. Please be more consistent with style. For example:

Line 42: Cluster 1 vs cluster 1 (capitalization or not)

Response: Thanks and we have ensure the style is consistent throughout the manuscript.

Line 179: Pseudomonas phages Pf1 vs. Line 184: Pseudomonas virus Pf1

Response: This has been updated to Pseudomonas virus Pf1. (Line 195 vs Line 202)

Figures: All p-values should be written consistently, probably with "p =" (not just a number), with same font size (e.g., Supp Fig 1), same bolding and italics or not (e.g., Supp Fig 4), same upper or lower case (e.g., Fig 3 legend), and the same format for exponentials (e.g., Supp Fig 3). In addition, many times the taxa are written with "_" instead of spaces. Please remove underscores.

Response: This has been done. We thank the reviewer for this comment. We have updated the figures for consistency. We have removed underscores of taxa.

3. Pathogens.

Line 46, 164, 215, 272: Are these really pathogens? I'm guessing maybe they've been shown to be opportunistic pathogens? If so, be sure to use the word opportunistic and cite where they've been shown to be pathogenic.

Response: We agree with the reviewer that these are opportunistic pathogens. We have changed to “opportunistic pathogens” with appropriate citation (Line 180, 239, 281).

Specific line comments:

Line 46: Aspergillus and C. albicans are spelled incorrectly here and some other places.

Response: We have corrected the spelling of Aspergillus and *C. albicans* throughout the manuscript.

Line 52 and 103: Is it important to show significant digits to the hundredth place?

Response: We have changed the significant digits to one decimal place. (Line 52, Line 108)

Line 72: The notion of fungal communities in the healthy human GI tract is controversial, as most fungi tend to be dead and only transiently present from saliva and consumed foods. I won't make a fuss about using the possibly technically incorrect term "mycobiome", but here perhaps just remove the word "communities" and say ... of viruses and fungi which...

Response: We have updated as proposed and removed the word “communities”.

Line 103: WSNF was defined, but not WSF.

Response: This has been done. (Line 103)

Line 109: MaAslin. I believe the L is also capitalized.

Response: We have corrected this to capitalized “L” (Line 115)

Line 119: CXCL10 has not been mentioned previously. What is it?

Response: Full-term “C–X–C motif chemokine 10” is provided (Line 124)

Line 146: ALP and ALT have not been defined previously.

Response: Full terms of ALP and ALP are provided: Alkaline phosphatase (ALP), Alanine transaminase (ALT) (Line 156).

Line 165: Why were the 2nd and 3rd most discriminant bacteria specifically mentioned? (ignoring #1)

Response: The 1st discriminant bacteria *Erysipelatoclostridium ramosum* has been included. (Line 180)

Line 172: Which time points are considered for this analysis? All?

Response: Baseline stool, blood, sputum and nasopharyngeal samples were considered for this analysis. We have clarified this in the manuscript. (Line 188)

Any thoughts on differences between Fig 4e-g and the taxa identified by MaAsLin?

Response: MaAsLin analysis and the random forest model use different data transformation methods and statistical assumptions, but they are complementary to each other. MaAsLin is a multivariate method, which identifies differential taxonomic biomarkers while accounting for the multiple

covariates. So, the biomarkers identified by MaAsLin have been adjusted by potential confounders and/or covariate included in the analysis such as demographic and clinical features. However, there is a possibility that demographic and clinical features could also predict the clinical outcomes of the patients. Therefore, in the random forest model, we included demographic and clinical features together with microbial taxa as predictors in order to achieve optimal prediction. The additionally clinical data and sub-samples of the training dataset used in the random forest model may also lead to the slight differences between random forest and MaAsLin.

Line 181: forest not forrest

Response: We have corrected this. (Line 199)

Line 181: Why were 11 factors chosen?

Did this perform better than the top 10, top 12, or other numbers?

Response: We aimed to determine the highest possible accuracy with the least number of factors. We performed the test from the top 5 to top 20 and found that using the top 11 achieved the best performance based on this model. This information has been included in the results (Line 197-198).

“We next evaluated the sub model performance from the top 5 to the top 20 and found that using the top 11 achieved best performance based on this model.”

Line 197: What value says what taxa contributed most to the model? Is it possible to add that information? Looking at Supp Fig 6, there is no indication that fungi and viruses contributed more than bacteria. Perhaps that value would also explain why the three specific taxa are pointed out on line 198/199. Also, the last part of this sentence does not make sense. It's odd to say these taxa could be considered after they were just considered.

Response: Thank you for the comments. We have rephrased this sentence for clarity and the top microbiome taxa were from all three kingdoms. We have made this clearer in the results (Line 216-219):

“The microbiome taxa that contributed most to the model to determine duration of viral shedding were from the three kingdom classes, including *Adlercreutzia equolifaciens*, *Asaccharobacter celatus*, *Candida dubliniensis*, *Klebsiella phage vB KpnP SU50*, and *Rhizobium phage vB RglS P106B* (Supplementary Figure 6).

Line 248: Remove "For example,"

Response: It has been removed. Thank you.

Line 349: One could make the case that most people have special eating habits. I'm surprised to see vegetarians excluded. What fraction of individuals were excluded due to dietary preferences?

Response: We thank the reviewer for this comment. In previous studies, subjects following a strict plant-based diet showed a decrease in gut microbiome diversity which was caused by an enrichment of microbes specifically involved in degrading plant carbohydrates but was not perceived as a negative health hallmark. In this work, only less than 3 individuals were excluded due to dietary preferences.

Line 406: How deep was the sequencing? If it's stated, I don't see it.

Response: The sequencing depth was an average of 6.9 Gbp per sample. We have added this information to the results section (Line 98)

Line 412: Maxwell RSC machine (what company?)

Response: It is from Promega. We have added this information to the methods section

Line 431 "Maxwell RSC machine (Promega, Wisconsin, USA)".

Line 415: I'm guessing the phrase "to obtain viral DNA, respectively" is not needed (what else would viral DNA kits be used for?)

Response: We have removed this phrase "to obtain viral DNA, respectively". (Line 433)

Line 430: I'm familiar with Metaphlan, but have never heard of MiCoP. This should either be explained or referenced.

Response: Micop [1] has been shown to identify more eukaryotes in human microbiome data than existing methods. We also found that MiCoP consistently identified a more diverse fungi than MethPhlAn3 across all samples. We have included this message and the reference to the method section (Line 449-450).

[1] LaPierre, N., Mangul, S., Alser, M., Mandric, I., Wu, N. C., Koslicki, D., & Eskin, E. (2019). MiCoP: microbial community profiling method for detecting viral and fungal organisms in metagenomic samples. *BMC genomics*, 20(5), 1-10.

Line 680/681: What is the y-axis for (C)?

Response: This volcano plot showed the spread of levels of function based on fold change and False Discovery Rate (FDR), representing statistical significance. We have included this to the figure legend (Line 724):

"Volcano plot showing the effect size log₂ fold-change (x axis) and significance (y axis) of the levels of function between the two clusters."

Specific figure comments:

Figure 2: (A) Adjust position of months, (C) Bifidobacterium (spelling)

Response: We have adjusted the position and corrected the spelling.

Here and elsewhere I find the color schemes confusing. It is fine that red is Cluster 1 throughout and blue is Cluster 2 throughout, but then it is confusing that bacteria are colored the same red, the mycobiome is colored the same blue, and the virome is colored the same green as the chi squared tests.

Response: We have adjusted the color schemes for bacteria, fungi, viruses, which are different from Cluster 1 and Cluster 2.

Figure 3: Fix spelling for fatigue, dizziness, and align taxa names better with bars

Response: We have corrected the spelling typo and align the taxa name better with bars.

Figure 4:

4B +C axis titles are bold and the others are not. Taxa names should be italicized.

4H. Needs more information than Positive Time. Not clear until read text or legend that referring to viral shedding.

Response: We have changed the font of axis title and taxa names are italicized. We have replaced "Positive Time" by "Viral shedding duration", which is consistent with the text in the manuscript.

Figure 5D: If the large central blue dot is *C. albicans*, where is the dot for *Wickerhamomyces*?

Response: We have adjusted the co-occurrence analyses to supplementary Figure 7. The dot for taxa is clearer.

Supp Fig 1: Definitions are needed of what the boxplots mean (line = median, etc).

The stats for B. should be moved from A to B.

Define the circles in B (centroids?)

Response: The line of boxplot indicates the median value. Box: quartile 1 to quartile 3; Vertical bar: minimum to maximum. Circles indicates the cluster centroid. We have added this to the figure legend. (Line 722-723)

I don't see any mention of Supp Fig 2C in the text.

Response: We have added the description of Supp Fig 2C in Line 127-128.

Supp Fig 3D: I doubt the p-value is truly 0. Perhaps put <0.01 ?

Response: This has been updated.

Supp Fig 5C+D: I was very confused why there is essentially no difference between Cluster 1 and 2. But now I think it is just a color scheme issue, as the legend says baseline (red like Cluster 1) and 6-month follow-up (blue like Cluster 2). Also, reading the legend I see that C = Cluster 1 and D = Cluster 2. This should be labeled on the figure, just like with A + B.

Response: We have changed the color pattern of baseline and 6 month follow-up which is different with the color of Cluster 1 and Cluster 2. The labels of C and D (C = Cluster 1 and D = Cluster 2) have been updated.

Reviewer #2 (Remarks to the Author):

Summary

Liu et al. present in their manuscript an analysis of COVID-19 patients with and without post-acute COVID-19 syndrome based on their gut microbiome profiles, and additional demographic and clinical data. While the herein included data and the described analysis are interesting and valuable, not least in the light of the ongoing pandemic, the manuscript often lacks important information as well as a more thorough discussion. In particular, the code and data required to reproduce the study are missing, and many passages in the Results and Methods sections are confusing (e.g. inconsistency in sample numbers and insufficient description of the analysis). Most importantly, the authors have already published a very similar study, which is also being cited in the present manuscript (<http://dx.doi.org/10.1136/gutjnl-2021-325989>, reference 10). However, the newly obtained results described here are not being discussed in the context of the previously published study.

Response: We thank this reviewer for the valuable comments and we have addressed them in the point-by-point response below.

Major comments

Comment 01

The authors demonstrate interesting correlations of the taxa with COVID-19 severity. However, aside from the single mention of the use of multi-omics for weighted similarity networks, the use of multi-omics is very limited. Therefore the title could be reassessed and revised. Importantly, given the availability of other COVID-19 datasets, it may be important to validate their in silico findings in a random publicly-available subset to ensure their robustness.

Response: Thank you for the suggestion. We have removed the word multi-omics and revised the title to “Multi-kingdom gut microbiota analyses define COVID-19 severity and post-acute COVID-19 syndrome”

We agree with the reviewer that it is important to test the robustness of the findings in publicly available subsets. However, after attempting we found that important metadata and sequencing data of published studies are not available, thus we were unable to include them in our analysis. We have included this as a limitation in the discussion (Line 320).

Comment 02

The post-acute COVID-19 syndrome (PACS) is defined as the presence of long-term complications observed after 4 weeks after the onset of symptoms (L338-340). This is a rather short time period and it is not clear from the manuscript whether these symptoms were reassessed at the 3 and 6 months time points to reevaluate their persistence.

Response: Thank you for the comments. In this study, the persistence of post-COVID symptoms were reassessed at 3 and 6 months. This has been included in the paper in Line 355-356: “We assessed the persistence of the 30 most commonly reported post-COVID symptoms at 3 and 6 months...”

Nalbandian, A., Sehgal, K., Gupta, A., Madhavan, M. V., McGroder, C., Stevens, J. S., & Wan, E. Y. (2021). Post-acute COVID-19 syndrome. *Nature medicine*, 27(4), 601-615.

Comment 03

Reproducibility.

The authors do not provide the information required to reproduce their study: there is no reference to a code repository to reproduce the described analysis; there is no raw data accession in the Data Availability statement and the one provided in the Methods section (i.e., PRJNA714459) is from another already published study; there is no sample metadata including the demographic information, laboratory measurements, PACS symptoms etc.; there is no processed data such as the microbiome profiles, built prediction models, and data used to generate the figures.

Response: Thank you for the comments. We have added the link to code availability and data availability in the paper.

Code availability: All bioinformatic and machine learning model scripts are available on Github (<https://github.com/g-micro/multi-omics>).

Data availability: Sample information and raw sequences are available in the National Center for Biotechnology Information Sequence Read Archive under BioProject

ID: PRJNA876804 (Supplementary Table 9) and will be made public upon formal publication. We have added a statement of this in the revised manuscript.

Comment 04

Sample numbers.

There seems to be some discrepancy in the number of samples and missing information on the sample size for some analyses. For example, the Abstract states 296 fecal metagenomes, however 293 is mentioned in L92. Furthermore, the authors describe the assessment of only 79 fecal samples for metabolomics (L94). It is unclear why or how this number was arrived upon. Additionally, it is not evident whether the profiles from 79 samples are representative of the 133 patients (or 296 fecal) samples collected. Similarly, the viral load was assessed only for 94 samples (upper and lower respiratory tract, and fecal samples, L373). There is also no information on the missing stool samples: there have to be $3 \times 133 = 399$ fecal samples in total so there are 103 to 106 missing samples. It is not stated whether the missing samples affect certain time points more than others.

A full list of the samples with the appropriate metadata should be provided for review. Any sample that was not used for each of the various assays should be clearly indicated with justification. In addition, statistical justification along with a stratification of the different patient/sample groups is required.

Response: We thank the reviewer for raising this important issue. We have provided the full list of samples for each analysis including metagenomic sequencing (at baseline, month 3 and month 6), blood cell count, plasma cytokines at baseline and medical records during hospitalization, PACS at month 3 and month 6. These data are included in Supplementary Table 9.

Comment 05

Found patient clusters and comparison to previous studies.

It is interesting to see two distinct clusters based on the similarity network approach. However, the description lacks some details. For example, it is not clear whether “pooling” in L101 is referring to the weighted similarity network fusion approach and whether “baseline” in L97 describes the time point “at admission”.

Response: Thank you for the comments. We used weighted similarity network fusion that takes into consideration the richness of the gut microbiome. For better clarity, we have modified the text to “By subjecting multi-biome data to a non-supervised similarity network fusion approach, fecal samples were divided into two distinct patient clusters based on their microbiota matrix.” (Line 106)

The baseline is the time point “at admission”. We have modified the text in the paper “Gut multi-biome (bacteria, fungi, virus) profile at admission was integrated...” (Line 102)

The authors state that the found two clusters had similar demographic parameters except for age: Cluster 1 contained older patients than Cluster 2 (L122-123). Given that the risk of higher disease severity for COVID-19 increases with age, it cannot be ruled out that the found clusters represent different age groups and that the observed differences in microbiome profiles and symptoms might be the consequence of that. The authors should consider this option and investigate whether patients’ age is correlated with their symptoms.

Response: Thank you for the valuable comments. We examined whether patients’ age were correlated with the post-COVID symptoms and found that there were no significant differences in the age of patients with PACS at six months. We have added this in the revised paper in Line 170-171

“Although older age is recognized in Cluster 1, there were no significant differences in the age of patients with PACS at six months between two clusters.”

For the time point “at admission”, there is also the possibility that the patients were admitted to the hospital at different stages of infection which might be also reflected in their microbiome and viral load. The authors should discuss this in the manuscript as it might also have contributed to the observed cluster formation.

Response: Thank you for the comments. During the early stage of the COVID-19 pandemic in Hong Kong, every confirmed case was admitted to the hospital on the same day or one day after positive SARS-COV-2. The time point “at admission” of the sample was the first stool sample collected after hospital admission so there is unlikely to be a big lag. Despite the timely admission to the hospital, there is still the possibility that patients were at different stages of infection. We have included this in the discussion. Line 305-307: “Besides, despite the timely hospital admission and sample collection, there is also the possibility that the patients were admitted at different stages of infection which might be reflected in their viral load and gut microbiome composition.”

Could there be a bias based on the geographic origins of the patients? It is also likely that the “severe patients” were housed in the same if not similar wards, compared to those with milder symptoms. Based on the Methods section, the possible biases induced due to housing conditions at the hospital (i.e. rooms) are unaccounted for. A reader would benefit from such information which can be contributing factors to microbiome profiles (see <https://doi.org/10.1128/msphere.01007-21>).

Response: Thank you for this point and highlighting the possible impact of geographic origins and housing conditions at the hospital.

Among the 133 patients, 110 patients were from Prince of Wales Hospital, 17 from United Christian Hospital, 6 from Yan Chai Hospital. Since most of the patients (110/130) were assigned to the same hospital, which is nearest to their geographic location in Hong Kong, the bias based on the geographic origins of patients should be limited in this study.

Based on the triage prioritisation in Hong Kong, only patients with critical COVID-19 could be provided ICU at admission, the rest of patients were housed in the same ward condition. Moreover, the baseline samples were collected on the day of hospitalization, limiting the bias based on the different wards. In this study, there were 12 critical patients who were admitted to ICU care and the

rest of the subjects had the same housing condition, limiting the impact of confounding factors. In addition, there was no statistical difference of the ratio of critical patients in the two clusters.

Finally, the authors state that the microbiome was stable over time within the clusters (L159-161). Furthermore, the majority of patients in Cluster 1 (84%) had PACS while only 44% were present in Cluster 2 (Fig. 3C, L166-168). However, in their other study, the authors also describe a recovery of the gut microbiome in patients without PACS at 6 months after disease resolution (<http://dx.doi.org/10.1136/gutjnl-2021-325989>, reference 10). Based on that, one would expect to see at least some temporal changes in Cluster 2 containing patients who did not develop PACS. Given this discrepancy, the authors should additionally compare the microbiome profiles (alpha- and beta-diversity) of patients with and without PACS and also across the different time points. In general, while the authors do cite their previous study in the present manuscript, they do not compare the newly obtained results to previous observations made in COVID-19 patients, though both studies have certain similarities in their design.

Response: We thank the reviewer for this comment. We analysed the gut microbiome profiles (diversity) of patients without PACS at 6 months in Cluster 2 and across different time points. The results showed a stable trend from baseline to 6-month follow-up (Supplementary Fig 5E and 5F) with or without PACS in patients.

The previous study solely focused on the bacterial microbiota, where gut microbiome in patients without PACS at 6 months after disease resolution, where patients exhibited a recovery gut microbiome composition. However, the key difference of the current analysis is the multi-kingdom microbiome analysis including include bacteria, fungi and viruses, which may explain the discrepancy between current and previous study.

We have included this in the paper Line 174-177:

“We further assessed whether there were temporal changes in patients without PACS in Cluster 2. The multi-microbiome exhibited stable microbiome profiles from baseline to as long as 6 months follow-up, indicating the persistent impact of SARS-COV-2 infection on the gut multi-kingdom microbiome.”

Comment 06

Microbiome profiling and data integration.

The authors state that the microbiome profiling was done to generate datasets representing three biomes: viruses (using a custom assembly-based approach), bacteria and fungi (using MetaPhlan3 and MiCoP). These data sets were then combined using weighted similarity network fusion (WSNF). There are multiple points which have to be clarified regarding this part of the analysis.

1) MetaPhlan3 includes in its database information bacterial, archaeal, viral, and eukaryotic genomes. Similarly, MiCoP estimates abundance of viral and fungal organisms. It has to be clarified whether the authors used all obtained hits from these two tools or if they limited the output only to specific organism groups, e.g. bacteria and fungi. If all hits were retained then it should be stated how the authors deduplicated the results to avoid multiple hits to the same species from different tools and approaches.

Response: Micop [1] has been shown to identify more eukaryotes in human microbiome data than previous-used method. We also found that MiCoP consistently identified a more diverse fungi than MethPhlAn3 across all samples.

For fungi, MetaPhlAn3 identified 2 genera in the samples, *Candida* and *Aspergillaceae*, while MiCoP identified 6 genera, including the two identified by MetaPhlAn3. Additionally, while the *Candida* genus dominated the MetaPhlAn2 results with 96.3% abundance, genera identified by MiCoP were distributed in a more balanced manner.

[1] LaPierre, N., Mangul, S., Alser, M., Mandric, I., Wu, N. C., Koslicki, D., & Eskin, E. (2019). MiCoP: microbial community profiling method for detecting viral and fungal organisms in metagenomic samples. *BMC genomics*, 20(5), 1-10.

2) The WSNF method assigns weights to the different biomes based on their richness: the number of species found in each biome. The number of observed species is highly affected by the used tools and their databases. Does the used approach account for that? Moreover, the authors should reference the respective publication describing the used WSNF approach.

Response: We thank the reviewer for this suggestion. Indeed, the richness will affect the similarity network infusion. In this study, the approach we used is to assume differential influences of each biome on the overall multi-biome based on both taxonomic composition and richness. Weighting was assigned according to the total number of observed species in a particular biome. We've cited the reference in the manuscript where we described the approach.

Line 102 "Gut multi-biome (bacteria, fungi, virus) profile at admission was integrated by an unsupervised weighted similarity network fusion (WSNF) approach⁴."

[4] Mac Aogáin M, Narayana JK, Tiew PY, Ali N, Yong VF, Jaggi TK, Lim AY, Keir HR, Dicker AJ, Thng KX, Tsang A. Integrative microbiomics in bronchiectasis exacerbations. *Nature medicine*. 2021 Apr;27(4):688-99.

3) The authors should state why they decided to pursue the assembly-based custom approach for virome profiling instead of using the output from MetaPhlAn3 and/or MiCoP.

Response: Identification of viral sequences in the process of viral metagenomic analysis is notoriously challenging due the lack of a universal viral marker as opposed to bacterial 16S rRNA for example. Thus, the reference-based read mapping is limited by a scarcity of annotated viral genomes. We used the optimized pipeline, capable of de novo extraction and retrieval of viral contigs from shotgun sequencing reads. Owing to the use of assembly and classification approach, we were able to retain only viral sequences for downstream analysis. We have included this in the paper (Line 454-458).

4) It is not completely clear from the text whether the integration of the three biome datasets was done using all available samples or for each time point separately (i.e., at admission, 3 months, and 6 months).

Response: Thanks for this comment. The integration of three biome dataset was performed on the admission stool samples. We next examined and compared the gut microbiome composition and occurrence of post-acute COVID-19 syndrome at different time points after acute infection. We have provided the samples list for integration of three biome datasets clearly.

Comment 07

L99/L457-458: The authors conveniently use a prevalence cutoff of 5% when filtering the dataset. However, it is unclear how they arrive at this number. Additionally it should be justified with appropriate references (see PMID 33510727). Also, it is not clear how the “average abundance of 1%” was used for filtering and why the authors chose the 1% cutoff. Moreover, the authors should state how many features were removed from the dataset using their filtering criteria and how many remained and were used for further analysis.

Response: We thank the reviewer for this comment. The data filtering procedure is applied to remove interactions resulting from random noise at the expense of sensitivity to weak signals. The cut-off value is suggested by the developer of weighted similarity network fusion for microbiome. Because of the high-coverage sequencing in this study, it is possible to detect some very-rare species with very low prevalence. We have cited the appropriate reference to the method section and the total number of species retrieved and remained for analysis (Line 486).

Reference: Mac Aogáin M, Narayana JK, Tiew PY, Ali N, Yong VF, Jaggi TK, Lim AY, Keir HR, Dicker AJ, Thng KX, Tsang A. Integrative microbiomics in bronchiectasis exacerbations. *Nature medicine*. 2021 Apr;27(4):688-99.

Comment 08

L137: The authors claim that blood urea levels were correlated with disease severity. However, based on the figures, a majority of the points are beyond the 2x standard deviation, suggesting a potential artifact due to outlier samples. This analysis has to be adjusted for outlier effects and assessed critically.

Response: We thank the reviewer for this comment. We have reassessed the correlation of blood urea levels and severity using Wilcoxon, which is a non-parametric test. These tests do not require a distributional assumption and are robust to the presence of outliers. Instead of transforming the data or discarding outliers, Wilcoxon Test is widely used to test whether there is difference in the values. We have updated the figure in supplementary Figure 1E.

Comment 09

L205: The authors claim an ‘ensemble-based’ method for the network analyses. However, the Methods section only describes using the “Reboot” approach. It must be noted that Reboot is quite outdated (2012) and since then additional/better methods for assessing sparse compositional matrices have been introduced. This is evident from Schwager et al.’s work (PMID: 29140991). It may be prudent to use an alternative method to corroborate the findings described herein.

Response: Thank you for the suggestion. We found sparse compositional matrices are more widely used. We have reassessed the network by using the widely used approach SparCC (Sparse Correlations for Compositional data). We have revised the content accordingly in results and methods (Line 222-247, Line 534-538)

Comment 10

L218: The authors demonstrate an example with *C. spiroforme*. However, it is unclear why this taxon was chosen. Especially given the fact that *R. gnavus* was also part of the description. It may be more prudent to use *R. gnavus* as an example to state the case, or explain why it did not show the expected correlations with the severe cluster.

Response: We agree it is more prudent to use *R. gnavus* as an example. We have made modifications in manuscript and figure (Line 244-246, Supplementary Figure 7F)

Comment 11

L226: It is surprising to observe that the “density” of the network is low, and yet the connected component value is 1. Could the authors explain the phenomenon driving this? In network theory, a highly connected component usually arises from a highly dense network, so this finding is counter-intuitive.

Response: In this section, we did not show the whole microbiome species, which may not be particularly informative and readable. We only showed those species with top co-occurrence values. The cut-offs of correlation coefficient and p-value are $|\text{Coef}| > 0.1$; $p < 0.05$, which led to the “low” “density” of the network. (Line 229)

Comment 12

For the patient stratification model, it seems that the clustering results obtained from the complete dataset were used as patient labels. If this is the case, then there might be an overfitting issue as the clusters and their optimal number were defined based on all available data. Therefore, the validation set used later does not completely represent truly unseen data as described by the authors (L472). A cleaner approach would be to perform the clustering using only the microbiome profiles of the training set. Expected labels for the validation set could be generated, for example, by a k-nearest neighbors approach to be able to compute the model accuracy.

Response: We thank the reviewer for this insightful comment.

1) In this section we used random forest model for stratification, which was different from the clustering approach. We would like to clarify the training dataset and testing dataset for model development. We have modified the description to avoid misunderstanding.

“Four datasets from 133 patients including demographic, blood test, cytokines and multi-biome were used separately or in combination (ensemble) to train the model. Machine learning models were first trained on the training set (70%, n=93) with five-fold cross validation, and then were applied to the test set (30%, n=40) for validation.”

2) We validate the clustering using only the microbiome of training set and five-fold cross-validation by two machine learning approach Support Vector Classifier (SVC,) and K-nearest neighbours algorithm. The AUROC yield above 0.9 indicating the robustness of clustering.

Another point is the feature ranking and selection procedure. Here it is not clear how exactly this was done (i.e., which data was used for feature selection, training and validation) and what the final model is. Additionally, the authors should not only look at accuracy but also consider other performance measures and their variance (in addition to the mean value).

The feature importance vector (mean decrease Gini index), including weights for every species predictive capacity, was collected. The Random Forest algorithm has built-in feature importance which can calculate the Gini importance (or mean decrease impurity) from the Random Forest structure. Mean decrease impurity" is defined as the total decrease in node impurity (weighted by the probability of reaching that node (which is approximated by the proportion of samples reaching that node)) averaged over all trees of the ensemble. The training dataset (70%) was used for feature selection.

We have clarified this in the paper in Line 504-507.

"The training dataset (70%) was used for feature selection. A trained forest produces a variable importance list based on mean decrease in Gini index. The feature importance vector (mean decrease Gini index), including weights for every species, demographic, blood test, or cytokines predictive capacity was collected."

The model was repeated ten times to obtain a distribution of random forests prediction evaluations. The final model was chosen looking at the best overall AUC value and overall accuracy. We have clarified this in the paper (Line 503, Line 524-526)

Similarly, the paragraph about the RF model used to predict the time period of patients being SARS-CoV-2 positive lacks some details to understand the performed model training and validation procedure. The authors should try to provide a more concise and clear description of this analysis step.

The dataset was divided into 70% training and 30% testing set. We have reorganized this paragraph to be clearer and more concise. (Line 519-520)

Minor comments

L204-205: It would be better to not introduce additional names or labels for Cluster 1 and Cluster 2.

Response: Thanks for the suggestion. We agree to keep the consistent label of Cluster 1 and Cluster 2 throughout the paper. We have removed the additional names of for Cluster 1 and Cluster 2.

L46: *R. gnavus* are commensal taxa found in many mammalian microbiomes. They are not pathogenic bacteria, unless in a particular context such as septic arthritis. Please refrain from using this misleading description

Response: We have edited the text. "was characterized by enriched abundance of bacteria (*Ruminococcus gnavus*, *Klebsiella quasipneumoniae*)" Line 48

L90-92: The text reads such that viral loads were checked in PBMCs. Please rephrase.

Response: The text is rephrased. Line 90-93

We assessed viral RNA levels in nasopharyngeal swabs and fecal samples using quantitative PCR with reverse transcription (RT-qPCR). We also assessed plasma cytokines and chemokines levels and leukocyte profiles from freshly isolated peripheral blood mononuclear cells (PBMCs).

L111-114: Please show the 95% and 99% confidence intervals and provide the PERMANOVA analyses results as a supplementary table with special emphasis on the F statistic.

Response: We have showed 99% confidence intervals and the PERMANOVA results are provide in Supplementary Table 2.

L130: Additional details regarding how the functional profiling was performed are needed.

Response: Details regarding how the functional profiling was performed is added to the text

Line 137-140

“For functional annotation, we applied the Human Microbiome Project Unified Metabolic Analysis Network 3 (HUMAN3) pipeline that maps reads to functionally annotated organism genomes and uses a translated search to align unmapped reads to UniRef90 protein clusters”.

L138: Additional details about how specific microbiome species were associated with elevated urea levels are needed.

Response: We have added and revised how specific microbiome species were associated with elevated urea levels in Line 147-154

“Next, we investigated how specific microbiome species were associated with elevated urea in severe COVID-19. The relative abundance of urea cycle pathway and K01940 in the urea cycle were significant higher in Cluster 1. Furthermore, we found a marked increase in K01940 (argininosuccinate synthase, the key enzyme in urea cycle pathway, Supplementary Figure 3B) in the severe cluster (Supplementary Figure 3C), which was predominantly driven by *Klebsiella* species such as *Klebsiella quasipneumonia*, *Klebsiella pneumoniae* and *Klebsiella variicola* (Supplementary Figure 3D) by comparing the subclasses pathway and microbial contributors (quantify gene presence and abundance in a species-stratified manner).”

L209-211: Please provide the correlation coefficient and p-value cutoffs in the results with respect to what is termed a “co-occurrence” and what is not.

Response: The cut-offs of correlation coefficient and p-value are $|Coef| > 0.1$; $p < 0.05$. (Line 229)

L344: It is unclear if all the patients were housed in the same hospital/quarantine facility. Please elaborate.

Response: Among the 133 patients, 110 patients were from Prince of Wales Hospital, 17 from United Christian Hospital, 6 from Yan Chai Hospital. We have added this information to the manuscript in Line 343-344.

L363-364: Why do the authors mention the antibiotic treatment if having no antibiotics therapy 6 months before, during and 6 months after the SARS-CoV-2 infection was one of the inclusion criteria (L344-346)?

Response: Since antibiotics would be a big confounder of gut microbiome analysis. We would like to exclude this impact on gut microbiome. In previous analyses, we found the impact of antibiotics could be persistent for 1 month after hospitalization. In this work, only patients without antibiotic therapy before at least 6 months, during and 6 months after acute infection of SARS-CoV-2 were eligible for analyses.

L379: Were the whole blood samples collected at admission?

Response: Yes, the whole blood samples were collected at admission.

L386-404: Which samples were analyzed here? All 293 stool samples?

Response: We analysed metabolomics of 79 faecal samples at admission.

We have clarified this in the paper in Line 406

“The quantification of fecal metabolites from 79 faecal samples at admission was performed by Metware Biotechnology Co., Ltd. (Wuhan, China).”

L432: The subsection title “Bioinformatic Viral Processing” is confusing. E.g., “Viral profiling of metagenomics data” or similar would be more appropriate.

Response: We thank the reviewer for this suggestion. We have revised the subtitle to “Viral profiling of metagenomics data”.

L437: Were the contigs deduplicated?

Response: The redundant sequences of contigs were removed by using CD-hit-EST. We have added the information to the method section in Line 471-472.

L448-450: How were the OTUs defined and generated from the reads mapped to contigs? Why was the normalization done by total number only and not also by the OTU sequence length?

Response: We thank the reviewer for this comment. We also normalized the OTUs by the OTU sequence length. We have included this in the paper in Line 476.

L464-465: Which software was used for patient clustering? What is the relationship between the eigengap method to choose the optimal number of clusters and the number of nearest neighbors?

Response: The clustering was conducted by weighted SNF (WSNF) using an online tool (<https://integrative-microbiomics.ntu.edu.sg>). The eigengap method and K nearest neighbors were two parallel approaches imbedded in WSNF for the determination of clusters. We have revised the sentence and added the appropriate citation. (Line 492-494)

L467: While the features used for model training are described in this paragraph the authors do not explicitly say what the response variable is.

Response: We have revised the description of the random forest method the response variable is clearly presented. Line 498-500 “Four datasets from 133 patients including demographic, blood test, cytokines and multi-microbiome were used separately or in combination (ensemble) to train the model for cluster stratification.”

L470-473: Although the training and validation datasets are described, the ‘test’ dataset is missing in the description. Please clarify.

Response: The data was split into training and testing data (7:3), and five-fold cross validation was applied in the training data set (30%). We have revised the description.

Line 493-496:

“Machine learning models were first trained on the training set (70%, n=93) with five-fold cross validation, and then were applied to the test set (30%, n=40) for validation. Each time a new feature was added to the model, the performance of the model was re-evaluated using the above training and validation set.”

L489-493: Is the result of the training procedure the “sparse model” or how exactly was that model generated after ranking the features?

Response : A sparse model consisting of the top 10 features was then validated on the validation set (30%, n=40). The accuracy of using top 10 (below) was lower than using all features for viral shedding duration. We have included this in the paper (Line 215-216).

L503-504: Which variables are described here? Why were categorical variables represented as numeric values (numbers and/or percentages)? Was the microbiome data normalized or transformed?

Response: The variables are demographic features including age, gender and presence of any comorbidities (Table 1). To quantify the disease severity, the numeric values were used for comparison. It is not the microbiome data. We have revised this in the paper Line 541-542.

“Continuous variables of demographic features were expressed in median (interquartile range) whereas categorical variables (disease severity, 5-point scale of severity) were presented as numbers.”

L538: USPTO application number is missing.

Response: US application no is 63/355,443

REVIEWERS' COMMENTS

Reviewer #1 (Remarks to the Author):

Thank you to the authors for doing an excellent job of responding to my specific comments and making those requested changes to the manuscript.

My only complaint is that I see no evidence that the authors attempted to address my general comment about how the grammar needs to be improved. Skimming over the text, I see for example that there are still multiple grammatical problems with the paragraph starting on line 277 (Multi-kingdom microbiome...)

My sympathies to the authors if English is not their primary language, but having this clearly written is important. The good news is that this should be a relatively quick issue to address, as the problems are numerous minor errors rather than large structural problems.

Reviewer #2 (Remarks to the Author):

General comments:

The authors have done a commendable job of revising the manuscript to address previously raised concerns. There are, however, a few minor points that need to be addressed and added to the manuscript.

Comments:

Comment 02: Please include a detailed description of the findings at 3 and 6 months, and not simply state that the results are available in the Supplementary file

Comment 03: The link to the GitHub page does not work.

Comment 11: Add the description of how the cut-offs were established in the text. Simply providing the cut-off values does not explain the rationale.

Minor comments:

Please add to the discussion the unlikelihood of geographic biases on the samples' microbiome profiles, as described in the rebuttal.

REVIEWERS' COMMENTS

Reviewer #1 (Remarks to the Author):

Thank you to the authors for doing an excellent job of responding to my specific comments and making those requested changes to the manuscript.

My only complaint is that I see no evidence that the authors attempted to address my general comment about how the grammar needs to be improved. Skimming over the text, I see for example that there are still multiple grammatical problems with the paragraph starting on line 277 (Multi-kingdom microbiome...)

My sympathies to the authors if English is not their primary language, but having this clearly written is important. The good news is that this should be a relatively quick issue to address, as the problems are numerous minor errors rather than large structural problems.

Response: We thank the reviewer for the positive comments. We have found a native English speaker to proofread the manuscript. We present a revised version of the manuscript based on this additional comment.

Reviewer #2 (Remarks to the Author):

General comments:

The authors have done a commendable job of revising the manuscript to address previously raised concerns. There are, however, a few minor points that need to be addressed and added to the manuscript.

Response: We thank the reviewer for the positive comments. We present a revised version of the manuscript based on these additional comments.

Comments:

Comment 02: Please include a detailed description of the findings at 3 and 6 months, and not simply state that the results are available in the Supplementary file

Response: We have included the findings at 3 and 6 months in the results section.

Line 170-175

“ For α -diversity based on the Shannon index, we found higher values in 3 months than in baseline samples but no significant increase in the diversity of the microbiota in 6 months (**Supplementary Figure 5A, 5B**). Within Cluster 1 and Cluster 2, there was no significant difference in the gut microbiome composition at admission and follow-up samples at 3 months and 6 months (**Supplementary Figure 5C, 5D**, $p > 0.05$) within each cluster suggesting the gut microbiome profile was stable over time.”

Comment 03: The link to the GitHub page does not work.

Response: We have corrected this. The link works now.

<https://github.com/g-micro/multi-omics>

Comment 11: Add the description of how the cut-offs were established in the text.

Simply providing the cut-off values does not explain the rationale.

Response: We have included this in the text. Line 544 “The correlation strength exclusion threshold was 0.1 using SparCC default setting. “

Minor comments:

Please add to the discussion the unlikelihood of geographic biases on the samples' microbiome profiles, as described in the rebuttal.

Response: We have included this in the discussion. Line 267-271 “Among the 133 patients, 110 were from the Prince of Wales Hospital, 17 from the United Christian Hospital, and 6 from Yan Chai Hospital. Since most (110/130) of the patients were assigned to the same hospital, which is nearest to their geographic location in Hong Kong, bias based on the geographic origins of patients should be limited in this study.”